# VideoSEAL: Mitigating Evidence Misalignment in Agentic Long Video Understanding by Decoupling Answer Authority

**Chenhao Qiu** [1] **Yechao Zhang** [2†] **Xin Luo** [1] **Shien Song** [1] **Xusheng Liu** [1]

## Abstract

Long video question answering requires locating sparse, time-scattered visual evidence within highly redundant content. Although current MLLMs perform well on short videos, long videos introduce long-horizon search and verification, which often necessitates multi-turn, agentic interaction. We show that existing LVU agents can exhibit *evidence misalignment*: they produce correct answers that are not supported by the retrieved or inspected evidence. To characterize this failure, we introduce two diagnostics (*temporal groundedness* and *semantic groundedness*) and use them to reveal two pressures that amplify misalignment: *prompt pressure* from shared-context saturation at inference time and *reward pressure* from outcome-only optimization during training. These findings point to a structural root cause: the *coupled agent* paradigm conflates long-horizon planning with answer authority. We therefore propose the *decoupled planner–inspector* framework, which separates planning from answer authority and gates final answering on pixel-level verification. Across four long-video benchmarks, our framework improves both answer accuracy and evidence alignment, achieving 55.1% on LVBench and 62.0% on LongVideoBench while producing interpretable search trajectories. Moreover, the decoupled architecture scales consistently with increased search budgets and supports plug-and-play upgrades of the MLLM backbone without retraining the planner. Code and models are available at https://github.com/Echochef/VideoSEAL.

[1]Mango TV, Changsha, China [2]Nanyang Technological University, Singapore. Correspondence to: Yechao Zhang <yech.zhang@gmail.com>.

*Proceedings of the 43rd International Conference on Machine Learning*, Seoul, South Korea. PMLR 306, 2026. Copyright 2026 by the author(s).

## 1. Introduction

Long-video understanding (LVU) is substantially more challenging than short-video reasoning because relevant evidence is sparse and temporally dispersed, while the video contains overwhelming irrelevant content (Lin et al., 2023; Cheng et al., 2024; Li et al., 2024). As a result, LVU systems are typically *agentic*: they iteratively navigate the timeline, retrieve candidate spans, and inspect visual evidence over multiple rounds of tool use (Wang et al., 2024; Ma et al., 2025; Tian et al., 2025; Yuan et al., 2025).

The prevailing paradigm in existing LVU systems is a unified agentic orchestration, where a monolithic planner controls retrieval, inspection tools within a single shared context window, reasons primarily over accumulated textual traces (e.g., captions and frame descriptions), and delivers the final answer (Yu et al., 2023; Wang et al., 2025b; Zhang et al., 2024a; Pang & Wang, 2025). However, we observe a recurring failure mode in this paradigm: agents exhibit premature commitment, outputting an answer even when their interaction trace cannot provide sufficient grounded support (Xiao et al., 2024; Sarkar & Etemad, 2025). We formalize this phenomenon as *evidence misalignment* by evaluating outcome correctness alongside trace groundedness, focusing on the pathological *correct but ungrounded* regime. To characterize this failure, we introduce two complementary diagnostics: *temporal groundedness*, which checks whether the agent accesses relevant temporal regions during interaction, and *semantic groundedness*, which uses an LLM judge to verify whether the final answer is logically supported by the tool outputs in the trace. Together, these diagnostics test whether an agent truly earns its answer from accessed evidence and help identify the root cause of misalignment.

Our diagnostics first reveal a systematic driver of evidence misalignment at inference time. During inference, as the agent's tool traces grow longer and noisier, the planner is increasingly prompted to be decisive without reliably locating and reconciling the relevant evidence (Liu et al., 2023; Bai et al., 2024), drifting from verification toward plausible completion (Menon et al., 2025). We term this pathology *prompt pressure*. Prompt pressure shifts behavior from *evidence seeking* to *evidence fitting*: when support is missing, ambiguous, or hard to locate in a long trace, the

planner resolves uncertainty by falling back to parametric priors and generic plausibility templates, producing answers that are weakly supported by the trace.

On the other hand, during training, while reinforcement learning (RL) offers a promising route to learning effective multi-turn interaction policies (Xie et al., 2025; Yang et al., 2025), agentic LVU is often equipped with *outcome-only* rewards because fine-grained temporal grounding labels are scarce (Krishna et al., 2017; Fu et al., 2024; Wu et al., 2024). This constitutes another systematic driver of evidence misalignment, since the reward primarily reflects answer correctness rather than evidence grounding, encouraging short-cut behaviors that improve outcomes without improving alignment. We refer to this incentive-driven pathology as *reward pressure*. Under reward pressure, agents can maximize return by relying on speculative completion instead of verifying evidence, leading to a widening hallucination gap (Skalse et al., 2022; Everitt et al., 2021).

We argue these limitations share a common structural root: the coupled agent paradigm, where evidence seeking, inspection, and answer generation are conflated within a single model operating over the same context. To address this, we propose a decoupled planner–inspector framework, which strategically separates these distinct roles. This design gives the LLM planner a sparse, structured state for long-horizon navigation. An MLLM inspector then verifies candidate evidence and holds exclusive answer authority through an inspector gate. By architecturally decoupling these responsibilities and leveraging GRPO (Shao et al., 2024), we mitigate prompt and reward pressures, effectively preventing unsupported answers driven by unverified tool outputs. Our decoupled framework achieves 55.1% answer accuracy and 62.0% accuracy on LongVideoBench (Wu et al., 2024) under a fixed budget, while exhibiting consistent gains with larger search budgets and stronger inspector backbones. Our contributions are threefold:

1. **Evidence Misalignment Diagnostics:** We formalize evidence misalignment in agentic VideoQA and introduce two diagnostics, *temporal groundedness* and *semantic groundedness*, which reveal two structural drivers: reward pressure and prompt pressure.

2. **Decoupled Planner–Inspector Framework:** We propose the decoupled planner–inspector framework that separates long-horizon planning from answer authority, with an inspector gate that produces an answer only when the inspected visual evidence is sufficient.

3. **Empirical Gains and Scaling:** Across four long-video benchmarks, our framework improves both answer accuracy and evidence alignment, and scales monotonically with larger search budgets and stronger inspector models, without retraining the planner.

## 2. Related Work

**MLLMs for LVU.** Recent Multimodal LLMs (Bai et al., 2025a; Liu et al., 2025a; Wang et al., 2025a) have demonstrated remarkable proficiency in short-horizon video understanding by scaling model capacity and instruction tuning. However, extending these architectures to long video encounters a structural bottleneck: visual token counts grow with video duration, whereas decisive evidence remains sparse. Standard approaches resort to aggressive downsampling or token compression, inevitably sacrificing critical evidence (Pang & Wang, 2025; Song et al., 2023; Zhu et al., 2025). This limitation (Pang & Wang, 2025; Wang et al., 2024; Ren et al., 2025) has necessitated a shift from single-pass inference to iterative, agentic reasoning paradigms that actively navigate the video timeline.

**Agentic Evidence Seeking and Reasoning.** To overcome the computational bottleneck of single-pass inference, recent work adopts iterative, evidence seeking agents that retrieve candidate temporal spans and reason over intermediate textual summaries across multiple turns (Wang et al., 2024; Liao et al., 2024; Shang et al., 2024; Zhang et al., 2025b). However, these agents often rely on a strong monolithic planner to manage long-horizon search, which incurs substantial inference cost. More fundamentally, they typically reason over lossy intermediate textual summaries rather than visual evidence, so inaccurate summaries can propagate into evidence misalignment when key spans are not retrieved.

Distinct from inference-time methods, complementary approaches aim to learn planning and tool-use behaviors directly via RL (Gao et al., 2025). Frameworks such as Video-MTR and LongVT (Xie et al., 2025; Yang et al., 2025) train a single model to iteratively acquire evidence with intrinsic visual grounding (Yao et al., 2023; Shen et al., 2023). However, these methods typically place search, accumulated tool outputs, and final answering in the same shared context. This design can encourage speculative completion, where the model compulsively reasons answers from superficial correlation.

## 3. Diagnosing Evidence Misalignment

In this section, we empirically document the evidence misalignment in agentic LVU, where an agent outputs an answer while its interaction trace provides insufficient support. Our analysis proceeds in three steps. First, we formalize agentic LVU as a multi-step interaction and introduce two diagnostics to measure trace groundedness (Section 3.1): (i) a temporal-overlap test using ground-truth intervals, and (ii) a trajectory-level semantic audit by LLM. Then, we use these diagnostics to illustrate how and why this misalignment occurs during training (Section 3.2) and at inference

(Section 3.3). Finally, we introduce a structural remedy by separating planning from answer authority (Section 3.4). Together, these steps connect trace-level symptoms to their causes and motivate a concrete structural fix.

### 3.1. Formalization and Diagnosis Setup

**Notation.** We model agentic long-video understanding as a sequential interaction process with evidence seeking actions. Given a video $\mathcal{V}$ and a query $q$, an agent policy $\pi$ interacts with the environment for at most $K$ rounds, using tool actions from an action space $\mathcal{U}$ to query, retrieve, or inspect video content. At each round $t$, the policy $\pi$ produces a rationale $r_t$ and an action $u_t \in \mathcal{U}$, based on the query $q$ and current interaction history $h_{t-1}$ ($h_0 = \emptyset$):

$$(r_t, u_t) \sim \pi(\cdot \mid h_{t-1}, q),$$

then receives an observation $o_t \leftarrow \mathrm{Env}(\mathcal{V}, u_t)$ from the environment and updates history as $h_t := h_{t-1} \oplus (r_t, u_t, o_t)$. The interaction terminates when the policy $\pi$ decides to stop or when the budget $K$ is reached. At the terminal round $T \leq K$, the agent emits an empty tool action $u_T = \emptyset$ and a rationale $r_T$ that details why it stops, along with the final answer $\hat{a}$ to the query $q$. Throughout the interaction trace $\xi := h_T$, at each round $t$ the agent accesses a set of video evidence $v_t := E(o_t)$ in the video. We summarize the overall process as $\pi : (\mathcal{V}, q) \mapsto (\hat{a}, \xi)$.

**Correctness *vs* Groundedness** We assess an agent along two axes: outcome correctness and trace groundedness. Let $a^*$ denote the ground-truth answer and define outcome correctness as $C \in \{0, 1\}$, where $C=1$ iff $\hat{a} = a^*$. Trace groundedness is a binary indicator $G \in \{0, 1\}$ that captures whether the agent's final answer $\hat{a}$ is justified by its interaction trajectory $\xi$, without relying on information absent from the trace. The pair $(C, G) \in \{0, 1\}^2$ partitions trajectories into four regimes: correct and grounded $(1, 1)$, correct but ungrounded $(1, 0)$, grounded but wrong $(0, 1)$, and ungrounded and wrong $(0, 0)$. Our focus is the *correct but ungrounded* regime $(C = 1, G = 0)$ that characterizes *evidence misalignment*: although the answer is right, it is not supported by the trace. This indicates that the agent bypasses actual visual verification, instead achieving correctness through speculative guessing, parametric priors, or superficial dataset shortcuts. Such behavior exposes the critical lack of verifiability and interpretability of the agent.

**Diagnostic I: Temporal Grounding.** We assess whether the agent accesses relevant temporal regions during interaction. For every retrieved span $\tau \in \mathcal{E}(\xi)$ and annotated interval $\tau^* \in \mathcal{E}^*$, where $\mathcal{E}^*$ denotes the ground-truth regions, we evaluate temporal groundedness as follows:

$$G_t := \mathbb{I}\left[\max_{\tau \in \mathcal{E}(\xi), \tau^* \in \mathcal{E}^*} \mathrm{tIoU}(\tau, \tau^*) \geq \gamma\right]. \quad (1)$$

where $\mathrm{tIoU}$ measures temporal intersection-over-union and $\gamma$ is the threshold (set to the CG-Bench training mean; $\gamma=0.05$), consistent with CG-Bench's loose overlap criterion (Chen et al., 2025; Mun et al., 2020).

To quantify how often agents give the correct answers without accessing the ground-truth evidence ($C=1, G_t=0$), we define the *temporal hallucination rate* as:

$$H_t := \mathbb{P}(G_t=0 \mid C=1) = \frac{\mathbb{E}[C(1-G_t)]}{\mathbb{E}[C]}. \quad (2)$$

A higher $H_t$ indicates that the agent achieves answer correctness frequently without accessing correct evidence, exposing a gap between outcome correctness and groundedness.

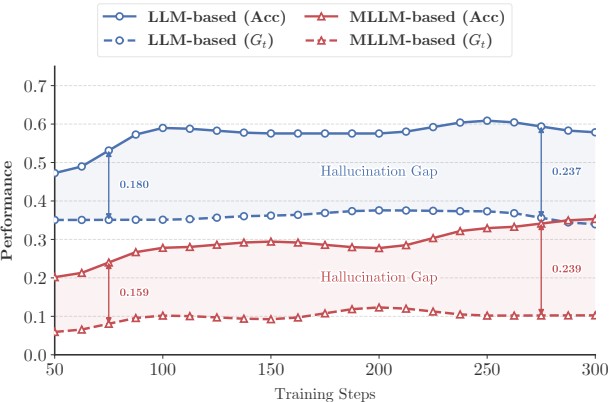

*Figure 1.* Performance vs Training steps. We train an MLLM planner (Video-MTR (Xie et al., 2025) with its original tools) and an LLM-planner (using the tools in Section 4.2) over CG-Bench.

**Diagnostic II: Semantic Grounding.** Temporal access alone does not guarantee grounded reasoning: even if the agent successfully retrieves the relevant spans for query $q$, it may not *properly interpret them and deduce the answer with sufficient reason.* To test whether an agent *earns* its answer from what it actually retrieves, we complement temporal overlap with an LLM-judge diagnostic that audits the interaction trace (Li et al., 2025). The LLM judge verifies the reasoning trajectory against tool outputs, where the answer lacking explicit grounded support is judged as a hallucination. Given $(q, \xi, \hat{a})$, the judge function $J_{\mathrm{judge}}$ checks whether all the claims that lead to $\hat{a}$ are unsupported by trace $\xi$. We define semantic groundedness as:

$$G_s := 1 - J_{\mathrm{judge}}(q, \xi, \hat{a}), \quad (3)$$

where $J_{\mathrm{judge}}=1$ signifies that the answer is unsupported (hallucinated). Analogous to Eq. 2, we quantify the *semantic hallucination rate* (correct but unsupported) as:

$$H_s := \mathbb{P}(G_s=0 \mid C=1). \quad (4)$$

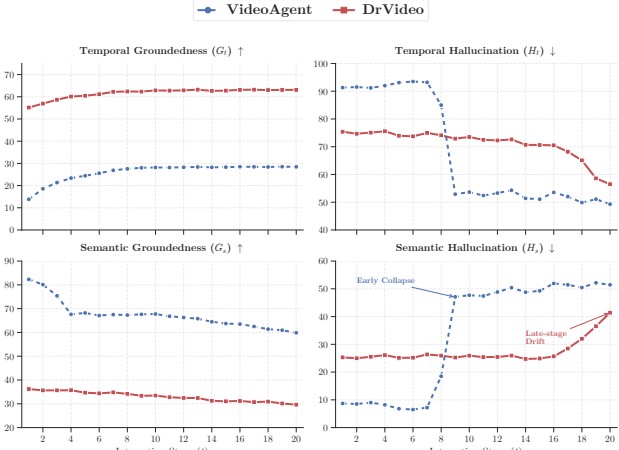

*Figure 2.* Temporal access *vs.* semantic support under trace growth. We report temporal groundedness/hallucination ($G_t/H_t$) and semantic groundedness/hallucination ($G_s/H_s$) for VideoAgent and DrVideo on LVBench.

While $G_t$ checks for temporal access, $G_s$ verifies logical semantic support, catching trajectory drift where the agent ignores accessed evidence.

### 3.2. Training-time Diagnostic: Reward Pressure

**Training longer cannot yield more grounded answers.** We observe *a systematic divergence between answer correctness and trace groundedness as training progresses* under the current prevalent agentic LVU solutions. Specifically, we evaluate across training steps using on-policy rollouts and compute both answer accuracy and temporal groundedness. Figure 1 shows the training dynamics across optimization steps: answer accuracy steadily improves for both LLM-based and MLLM-based agents, whereas trace groundedness grows much more slowly, leading to an expanding gap between outcomes and groundedness. This growing *outcome–grounding gap* indicates that correctness is not accompanied by proportional gains in evidence seeking. Instead, agents increasingly rely on parametric priors or spurious cues rather than retrieved evidence, leading to more correct yet ungrounded answers.

**Reward pressure impairs groundedness.** The above results suggest that training implicitly prioritizes outcome optimization over trace grounding. In the prevalent architectures, the planner is responsible for both interaction decisions and final answer generation. While reward signals directly supervise final answers, they provide only weak and indirect feedback on the evidence seeking process. Subsequently, agents can maximize rewards more efficiently by exploiting parametric priors or superficial patterns in the interaction trace, rather than performing costly and precise evidence retrieval (Chai et al., 2024; Cao et al., 2025; Light-

man et al., 2023). We refer to this training-time bias toward outcome-driven shortcuts as *reward pressure*. Under reward pressure, speculative reasoning that guesses answers using priors becomes more effective than rigorous evidence seeking, thereby progressively weakening trace groundedness and enlarging the outcome–grounding gap.

### 3.3. Inference-time Diagnostic: Prompt Pressure

**Seeking longer (still) cannot yield more grounded answers.** We observe a systematic failure pattern across different interaction trajectories: *longer interaction histories with more retrievals do* not *reliably yield more semantically grounded answers.* Specifically, we group interaction traces by their terminal length and compute both groundedness ($G_t$ and $G_s$) and hallucination metrics ($H_t$ and $H_s$).

As shown in Figure 2, temporal access and semantic support increasingly decouple as trace length grows: $G_t$ saturates early, while $G_s$ steadily declines. Temporal groundedness improves mainly in the first few steps and then plateaus for both VideoAgent and DrVideo, suggesting diminishing returns from additional retrieval or inspection. In contrast, semantic groundedness degrades monotonically with longer traces, accompanied by rising semantic hallucination. Overall, these reveal that longer interactions do not translate into better trace-supported reasoning. Instead, they increase the prevalence of *correct yet ungrounded* outputs: agents may touch relevant content, but their final answers are progressively less justified with sufficient reason.

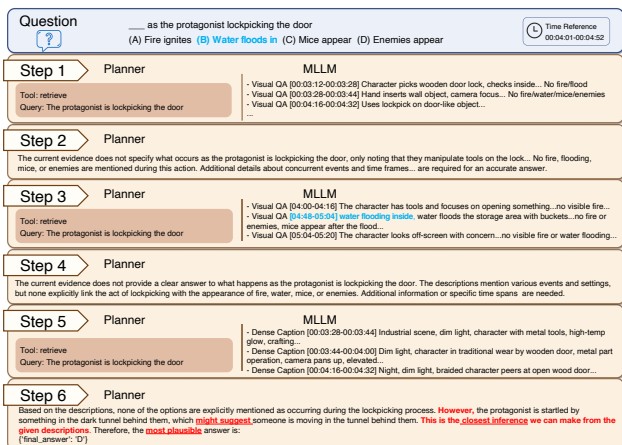

*Figure 3.* An example of prompt pressure. The agent repeatedly retrieves candidate clips and even surfaces potentially relevant visual cues, yet its final decision is a hedged plausibility template (e.g., "might suggest") and incorrect.

**Prompt pressure impairs groundedness.** The above results suggest that the agent is increasingly *prompted to be decisive without adequately inspecting retrieved evidence, regardless of whether the trace suffices to justify an answer.* In the prevalent architectures, the planner is also responsi-

ble for both trace inspection and final answer generation. A longer trace creates an incentive to commit even when trace support is uncertain. We refer to this inference-time compulsion to commit as *prompt pressure*. As showcased in Figure 3, prompt pressure shifts behavior from *evidence seeking* to *evidence fitting*: when support is missing, ambiguous, or hard-to-locate in a long trace, the agent resolves uncertainty by falling back to superficial correlation and generic plausibility, yielding seemingly coherent but weakly supported answers. Thus, the agent may still *touch* relevant content (so $G_t$ saturates), while its final decision drifts toward tentative guesses rather than grounded understanding, degrading $G_s$ and increasing semantic hallucination as $T$ grows.

### 3.4. Remedy: Separating Planning and Answering

**Structural root: the coupled agent paradigm.** Building on the above diagnostics, we attribute both *prompt pressure* and *reward pressure* to a shared structural cause: the *coupled-agent* design. We define a *coupled agent* as a single policy that both seeks evidence and produces the final answer, with all decisions conditioned on the same interaction history $h_t$. In prevalent architectures, this means the planner is simultaneously responsible for trace inspection, termination, and final answer generation within one shared context. Such entanglement amplifies *prompt pressure* at inference time and *reward pressure* during training.

**Decoupled agent with an inspector gate.** To break this entanglement, we propose a decoupled agent with two entities (see Figure 4): a planner that gathers evidence via tools, and an inspector that controls answer authority. After each retrieval/inspection, the planner submits inspectable evidence to the inspector, who returns a binary sufficiency verdict. The planner terminates only when the inspector approves. Otherwise, it continues evidence seeking.

## 4. Method

In this section, we present our decoupled framework in Section 4.1 and the tool set used for the planner's agentic evidence seeking in Section 4.2, as well as how the planner and inspector interact using the tool set in Section 4.3. Finally, we describe the reward design and policy optimization procedure for training the planner in Section 4.4.

### 4.1. Decoupled Planner–Inspector Framework

We propose a novel decoupled planner–inspector framework for agentic LVU. The key design is to decouple the original policy $\pi$ into two entities: a planner $P$ for evidence seeking and an inspector $I$ that exclusively controls termination and the final answer. Concretely, at each round $t$, the planner produces a rationale–action pair $(r_t, u_t)$ conditioned on the

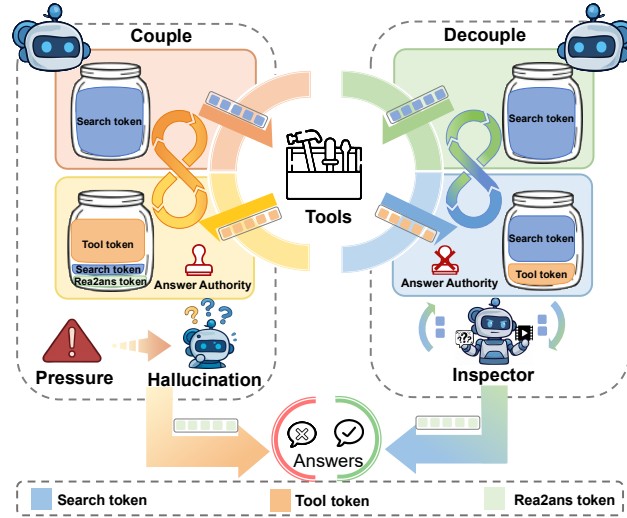

*Figure 4.* Architectural comparison between the coupled agent and the decoupled agent.

history $h_{t-1}$, the environment returns observation $o_t$, and the inspector evaluates the extracted evidence $v_t := E(o_t)$:

$$(r_t, u_t) \sim P(\cdot \mid h_{t-1}, q), \qquad (z_t, f_t) \sim I(\cdot \mid v_t, q).$$

Here $z_t \in \{0, 1\}$ is a binary sufficiency verdict and $f_t$ represents the feedback, where $z_t=1$ indicates that the current evidence is sufficient to answer. The agent terminates and outputs an answer only when $z_t=1$. Otherwise, the planner continues evidence seeking. Figure 5 illustrates the workflow, and we detail the planner–inspector interaction protocol in Section 4.3.

### 4.2. Tool Interfaces

We implement long-horizon evidence seeking with three tool interfaces: (i) offline clip indexing, (ii) index-based retrieval, and (iii) visual inspection.

**Offline indexing.** We partition each long video into fixed-length clips (16 seconds). For each clip, we pre-generate a caption using an MLLM captioner (Qwen3-VL-8B-Instruct) and a dense embedding of that caption using a text embedding encoder (text-embedding-3-large), and store the metadata in a retrieval index.

**Retrieval.** Given a text query, `VisualRetrieve` performs cosine-similarity search over the index to return a top-$k$ set of candidate clips/spans. To reduce semantic drift, we apply an LLM filter (DeepSeek-V3.2) over the associated captions to prune and consolidate candidates, yielding a concise set of question-relevant temporal spans.

**Visual inspection.** `VisualInspect` is the tool interface used to instantiate the inspector $I$. Given a set of video evidence $v_t$ and the query $q$, `VisualInspect`$(v_t, q)$ returns a structured response $(z_t, f_t)$. When $z_t = 1$, $f_t$ includes an

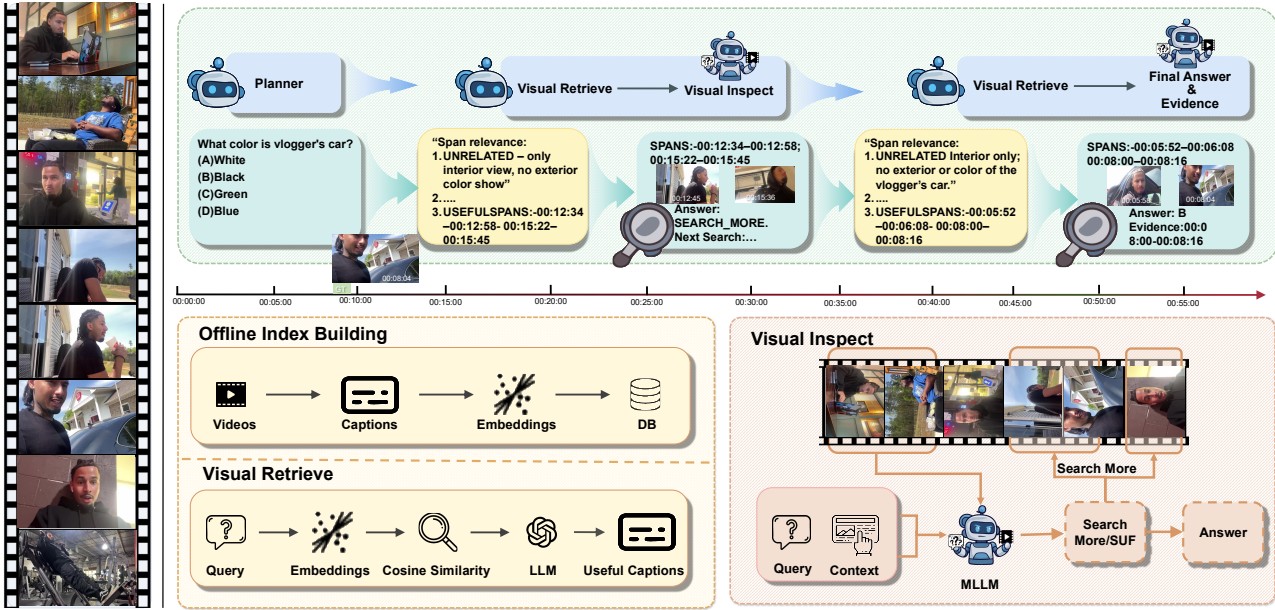

*Figure 5.* Overview of the decoupled planner–inspector framework.

answer proposal $\hat{a}_t$ grounded in the inspected evidence.

## 4.3. Planner–Inspector Interaction

**Planner.** The planner is a language-model policy for long-horizon evidence seeking. At each turn $t$, it takes the question $q$ and a compact search memory $h_{t-1}$, and chooses one of two actions: (i) retrieve candidate temporal spans, or (ii) submit a set of video evidence $v_t$ for pixel-level inspection via `VisualInspect`$(v_t, q)$. After an inspection, the planner appends the submitted spans and the inspector's feedback to its search memory, keeping the planner focused on evidence seeking rather than speculative answering. Crucially, this feedback closes the loop between verification and search: based on the inspector's verdict and feedback, the planner iteratively revises its span proposals, either broadening retrieval to explore new evidence or refining $v_t$ into a tighter, more informative window for re-inspection.

**Inspector.** The inspector is a frozen MLLM model accessed through `VisualInspect`. Each call is conditioned only on the original question $q$ and the submitted video evidence $v_t$, without access to the planner's intermediate reasoning or full history. This design reduces the influence of accumulated context and makes verification depend on what is actually visible in the inspected spans. As a result, the inspector controls termination and outputs the final answer only when the inspected evidence is sufficient. Finally, because the inspector is a modular visual backend, it can be upgraded at inference time by swapping in a stronger MLLM without retraining the planner.

## 4.4. Policy Optimization and Reward Design

**Optimizing the planner with a frozen inspector.** We optimize the planner $P$ using GRPO (Shao et al., 2024) while keeping the inspector $I$ frozen. This choice ensures that policy optimization primarily shapes long-horizon evidence seeking behavior (e.g., when to retrieve, which spans to inspect), rather than altering visual verification or answer generation. To disentangle the effect of architectural decoupling from reward specification, we evaluate both the coupled baseline and our decoupled framework under two terminal-reward regimes.

**Outcome-only terminal reward.** As a default baseline, we use an *outcome-only* terminal reward that depends solely on answer correctness. Let $a^*$ denote the ground-truth answer and $\hat{a}$ be the terminal prediction. We define

$$R_{\text{ans}}(\xi) := \begin{cases} 1, & \text{if } \hat{a} = a^*, \\ 0, & \text{otherwise,} \end{cases} \qquad (5)$$

which ignores how the answer is obtained.

**Evidence-gated terminal reward.** On datasets with ground-truth evidence intervals, we additionally encourage the policy to *access* temporally aligned evidence. Let $\mathcal{E}^*$ be the set of ground-truth evidence intervals, and let $(\xi)$ denote the set of evidence spans accessed by the trajectory (defined in Section 3.1). In practice, the maximum temporal overlap between accessed spans and ground-truth evidence is small (e.g., $\approx 0.05$ on average), making a hard evidence gate overly sparse. We therefore use a *bounded soft gate*

*Table 1.* Accuracy on four LVU benchmarks. For each agentic solution, we report the planner, inspector, parameter size, answer authority, and frame budget. In this table, Frames denotes the maximum number of frames fed into the inspector per inspection call.

| Models | Planner | Inspector | Param. | Ans. Auth. | Frames | MLVU | VideoMME (w/o sub) Overall | Long | LongVideoBench | LVBench |
|---|---|---|---|---|---|---|---|---|---|---|
| *Duration* | | | | | | 3∼120 min | 1∼60 min | 30∼60 min | 0∼60 min | 4101 sec |
| *Proprietary Models* | | | | | | | | | | |
| Gemini-1.5-Pro | – | Gemini-1.5-Pro | – | Model | 1fps | 61.8 | 75.0 | 67.4 | 64.0 | 33.1 |
| GPT-4o | – | GPT-4o | – | Model | 1fps | 66.2 | 77.2 | 72.1 | 66.7 | 34.7 |
| *Open-Source Video MLLMs* | | | | | | | | | | |
| Video-LLaVA (Lin et al., 2023) | – | Video-LLaVA | 7B | Model | 8 | 47.3 | 40.4 | 38.1 | 39.1 | - |
| LongVA (Zhang et al., 2024b) | – | LongVA | 7B | Model | 128 | 56.3 | 54.3 | 47.6 | - | - |
| Video-R1 (Feng et al., 2025) | – | Video-R1 | 7B | Model | 32 | 45.4 | 59.3 | - | - | 35.9 |
| Qwen2.5-VL-Instruct (Bai et al., 2025b) | – | Qwen2.5-VL | 7B | Model | 64 | 63.9 | 58.4 | 44.9 | 55.3 | 34.6 |
| Qwen2.5-VL-Instruct (Bai et al., 2025b) | – | Qwen2.5-VL | 7B | Model | 768 | - | 63.5 | 50.3 | 56.0 | 39.4 |
| *Agentic frameworks (Coupled)* | | | | | | | | | | |
| VideoAgent (Wang et al., 2024) | GPT-4o | – | – | LLM | – | 55.8 | 59.4 | 47.9 | 50.3 | 42.3 |
| DrVideo (Ma et al., 2025) | GPT-4o | Qwen2.5-VL | 7B | LLM | – | 62.4 | 62.5 | 50.4 | 55.9 | 51.4 |
| Ego-R1 (Tian et al., 2025) | Qwen2.5 | Qwen2.5-VL | 7B | LLM | – | 40.9 | - | - | - | 41.0 |
| GenS (Yao et al., 2025) | Qwen2.5-VL (3B) | Qwen2.5-VL | 7B | MLLM | 64 | 58.0 | 58.1 | 49.7 | 41.6 | 41.3 |
| FrameThinker (He et al., 2025) | Qwen2.5-VL | Qwen2.5-VL | 7B | MLLM | 21.1 | 59.1 | - | 47.6 | 52.9 | 36.6 |
| Conan (Ouyang et al., 2025) | Qwen2.5-VL | Qwen2.5-VL | 7B | MLLM | 32 | 63.4 | 60.5 | - | 56.6 | 39.2 |
| VideoMind (Liu et al., 2025b) | Qwen2-VL | Qwen2-VL | 7B | MLLM | 32 | 64.4 | 57.6 | 49.3 | 55.7 | 42.4 |
| Video-MTR (Xie et al., 2025) | Qwen2.5-VL | Qwen2.5-VL | 7B | MLLM | 80 | 58.4 | 62.7 | 52.7 | 57.3 | 42.0 |
| LongVT (Yang et al., 2025) | Qwen2.5-VL | Qwen2.5-VL | 7B | MLLM | 512 | 58.0 | 58.1 | 49.7 | 41.6 | 41.3 |
| Ours | Qwen3 (8B) | Qwen2.5-VL | 7B | LLM | 64 | 64.6 | 59.9 | 49.6 | 52.2 | 48.2 |
| *Agentic frameworks (Decoupled)* | | | | | | | | | | |
| Ours | Qwen3 (8B) | Qwen2.5-VL | 7B | MLLM | 64 | **68.2** (↑ 4.3) | **62.9** (↑ 4.5) | **53.4** (↑ 8.5) | **62.0** (↑ 6.7) | **55.1** (↑ 20.5) |

that scales smoothly with overlap:

$$g_{\text{evd}}(\xi) := \min\left\{1, \frac{1}{\gamma} \max_{\tau \in \mathcal{E}(\xi)} \max_{\tau^* \in \mathcal{E}^*} \text{tIoU}(\tau, \tau^*)\right\}, \quad (6)$$

where $\text{tIoU}(\cdot, \cdot)$ is the temporal IoU (Section 3.1), and $\gamma$ is an overlap scale (e.g., the training-set mean of the max-tIoU statistic) that normalizes the gate to $[0, 1]$. The evidence-gated terminal reward is then

$$R_{\text{evd}}(\xi) := R_{\text{ans}}(\xi) \cdot g_{\text{evd}}(\xi). \quad (7)$$

This reward remains outcome-correctness driven, but assigns higher return to trajectories that answer correctly *and* access evidence that better aligns with annotated ground-truth intervals.

# 5. Experiments

## 5.1. Experimental Setup

We evaluate on four LVU benchmarks: VideoMME (Fu et al., 2024), MLVU (Zhou et al., 2025), LongVideoBench (Wu et al., 2024), and LVBench (Zhang et al., 2025a). These datasets span video durations from minutes to hours and require multi-hop temporal reasoning.

To isolate the effect of architectural decoupling, we compare a coupled baseline with our decoupled framework using identical backbones and training objective (GRPO (Shao et al., 2024)). The only difference is answer authority: in the coupled baseline, the planner produces the final answer; in our decoupled framework, the inspector produces the final answer and decides when to stop. We also report comparisons with prior agentic frameworks under their standard

evaluation protocols, using the same backbones and search budgets where applicable.

## 5.2. Main Results

Table 1 reports results on four LVU benchmarks across (i) proprietary models, (ii) open-source video MLLMs, and (iii) agentic frameworks (coupled and decoupled). Within the same backbone, our decoupled framework consistently improves over its coupled counterpart. For example, it improves MLVU accuracy from 64.6% to 68.2%, and yields similar gains on LongVideoBench from 52.2% to 62.0% and LVBench from 48.2% to 55.1%. This indicates that separating termination and answer authority can translate evidence seeking into reliable gains.

*Table 2.* Grounding evaluation on LVBench. VideoSEAL improves temporal and semantic grounding beyond final answer accuracy.

| Method | Recall@0.05 ($G_t$) | Recall@0.10 | Recall@0.20 | $H_t \downarrow$ | $G_s \uparrow$ | $H_s \downarrow$ |
|---|---|---|---|---|---|---|
| LongVT | 0.088 | 0.054 | 0.041 | 0.889 | 0.250 | 0.708 |
| Video-MTR | 0.175 | 0.138 | 0.088 | 0.833 | 0.280 | 0.621 |
| VideoMind | 0.275 | 0.219 | 0.144 | 0.629 | 0.273 | 0.543 |
| VideoAgent | 0.281 | 0.231 | 0.169 | 0.838 | 0.454 | 0.471 |
| DrVideo | 0.448 | 0.324 | 0.271 | 0.547 | 0.472 | 0.414 |
| Ego-R1 | 0.419 | 0.325 | 0.275 | 0.504 | 0.405 | 0.378 |
| **Ours** | **0.528** | **0.434** | **0.333** | **0.406** | **0.808** | **0.113** |

Beyond final answer accuracy, Table 2 further evaluates whether the retrieved and inspected evidence actually supports the answer on LVBench. VideoSEAL achieves the best temporal grounding, with the highest Recall@0.05 ($G_t$), Recall@0.10, and Recall@0.20, and also substantially reduces temporal hallucination $H_t$. More importantly, it improves semantic groundedness $G_s$ to 0.808 and reduces semantic hallucination $H_s$ to 0.113, indicating that our decoupled

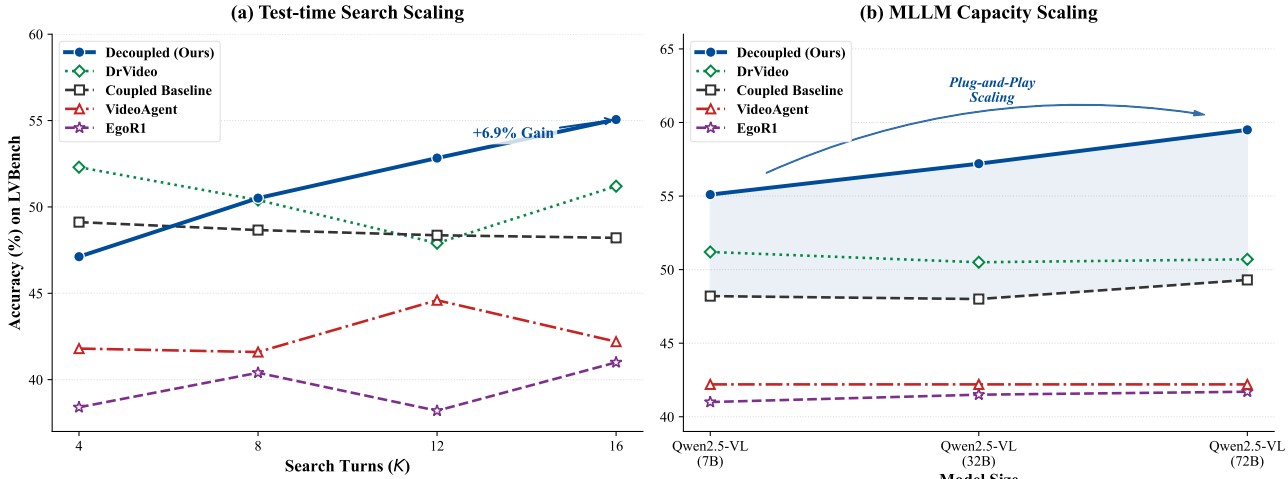

*Figure 6.* Scaling properties. (a) Search: Increasing budget $K$ improves our method, while coupled baselines plateau due to context saturation. (b) Perception: Replacing the inspector from 7B to 72B yields substantial accuracy gains without retraining the planner, highlighting modular scalability.

framework produces fewer correct-but-unsupported answers and yields more evidence-aligned trajectories.

*Table 3.* Decoupling dominates reward design. Decoupling consistently outperforms coupled baselines under both reward settings, identifying architectural separation as the primary performance driver.

| Model | Objective | Acc (%) | $G_t$ (%) | $H_t$ (%) |
|-------|-----------|---------|-----------|-----------|
| Coupled | $R_{\mathrm{ans}}$ | 48.2 | 47.0 | 48.3 |
| Decoupled | $R_{\mathrm{ans}}$ | 54.1 | 50.4 | 42.1 |
| Coupled | $R_{\mathrm{evd}}$ | 50.2 | 50.2 | 43.2 |
| Decoupled | $R_{\mathrm{evd}}$ | **55.1** | **52.8** | **40.6** |

## 5.3. Ablation Studies

**Decoupling drives the main gain.** To study the impact of the decoupling mechanism itself and identify the main source of improvement, we report a controlled $2 \times 2$ comparison over architecture and reward design in Table 3. Across all variants, the planner, inspector backbone, retrieval stack, and GRPO training details are kept fixed. The only architectural difference is whether stop and final-answer authority stays with the planner or is delegated to the inspector gate. We also compare the outcome-only reward $R_{\mathrm{ans}}$ with the temporal evidence-gated reward $R_{\mathrm{evd}}$. The results indicate that the gain is primarily due to decoupling, rather than auxiliary modules or reward enhancement. Although the time reward design helps, the coupled agent with $R_{\mathrm{evd}}$ still underperforms the decoupled agent with $R_{\mathrm{ans}}$ (50.2% vs. 54.1%). This suggests that the main improvement does not come from temporal reward supervision, but from architectural decoupling, which provides a structural remedy by separating planning from answer authority.

**Sequential Search Scaling.** We quantify the decoupled agent's ability to convert computational budget into performance by scaling the maximum number of search rounds $K$. As shown in Figure 6(a), increasing interaction steps improves performance from 47.1% to 55.1%, suggesting that investing more computation in exploration yields higher gains without increasing contextual burden. In contrast, when interaction rounds exceed 8, the coupled agent regresses, consistent with contextual saturation where accumulated history weakens reasoning reliability, whereas our framework remains robust and continues to benefit from larger budgets.

**Decoupled Perceptual Scaling.** Beyond search scaling, our decoupled agent naturally facilitates MLLM scaling. By structurally isolating answer authority from the planner to the inspector, we can directly replace the inspector without retraining the planner. As shown in Figure 6(b), scaling the inspector within Qwen2.5-VL from 7B to 72B yields a clear improvement on LVBench, increasing performance from 55.1% to 59.5%. This trend suggests that the decoupled agent is not tied to a specific visual model and supports plug-and-play scaling via a stronger MLLM backend, whereas the coupled agent shows only a weak 1.1% gain despite a large increase in parameters, indicating performance saturation.

*Table 4.* Refusal and answer rates on non-target vs. ground-truth clips. Refusal corresponds to $z=0$ and answer to $z=1$. We also report accuracy on answered cases $\mathrm{Acc}_{\mathrm{ans}}$ and the base accuracy of each inspector.

| Inspector Model | Clip Interval | $z=0$ (%) | $z=1$ (%) | $\mathrm{Acc}_{\mathrm{ans}}$ (%) | Base Acc. (%) |
|-----------------|---------------|-----------|-----------|-----------------------------------|---------------|
| Qwen2.5-VL-7B-Instruct | Non-target | 75.8 | 24.2 | 61.1 | 39.4 |
| | Ground-truth | 21.6 | 78.4 | 72.7 | |
| Qwen3-VL-8B-Instruct | Non-target | **85.4** | 14.6 | 53.5 | 50.6 |
| | Ground-truth | 26.5 | 73.5 | **84.0** | |

**Refusal Enables Selective Answering.** A key question for the decoupled framework is whether the inspector can recognize insufficient evidence and refuse to answer, instead of producing a plausible guess. To isolate this behavior on LVBench, for each question with annotated evidence timestamps, we construct two inspection inputs: (i) a randomly sampled 16-second *non-target* clip that does not overlap the annotated evidence, and (ii) a 16-second *ground-truth* clip sampled from the annotated evidence interval. We feed the visual content of each clip to the inspector and record its verdict $z \in \{0, 1\}$, where $z=0$ indicates a refusal.

Table 4 shows that even the Qwen2.5-VL base inspector exhibits selective answering. Specifically, the answer rate $z=1$ is substantially higher on ground-truth clips at 78.4% than on non-target clips at 24.2%, indicating that the inspector can distinguish evidence-sufficient from insufficient inputs. Conditioned on answering, accuracy is also higher on ground-truth clips at 72.7% than on non-target clips at 61.1%, suggesting that the decision to answer correlates with evidence quality. When scaling to a stronger inspector, we observe concurrent gains in both answered accuracy and refusal: answered accuracy on ground-truth clips increases to 84.0%, while refusal on non-target clips becomes stricter, with $z=0$ rising to 85.4%. Overall, these results suggest that the inspector is more likely to refuse on non-target clips, reducing guessing and encouraging the planner to keep searching for better evidence.

## 6. Conclusion

In this work, we analyze evidence misalignment as a reward-hacking failure in agentic long-video question answering, where agents can produce correct but insufficiently supported answers by optimizing answer correctness without grounding their decisions in verified evidence. We introduce temporal and semantic groundedness diagnostics to identify correct-but-ungrounded answers and measure temporal evidence access, and show that misalignment is amplified by prompt pressure at inference time and reward pressure during training. To address this, we propose a decoupled planner–inspector framework that separates long-horizon evidence seeking from answer authority and gates final answering through visual verification. Across four long-video benchmarks, our framework improves over coupled baselines, scales with larger search budgets, and supports plug-and-play inspector upgrades without retraining the planner.

## Impact Statement

This work frames evidence misalignment as a manifestation of reward hacking in agentic long-video QA: agents may optimize answer correctness while bypassing the intended process of evidence-grounded verification. We argue

for a simple shift in how we build agentic long-video QA frameworks: instead of training a single coupled agent to both search and decide, we separate evidence seeking from answer authority with a planner–inspector design. Alongside this decoupling, we introduce evidence-grounded diagnostics that make it easy to tell whether an answer is actually supported by what the agent retrieved. We hope this paradigm and analysis help the community build more verifiable long-video agents and reduce unsupported guessing.

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

# A. Experimental Details

**Training Data**   We train our planner on CG-Bench (Chen et al., 2025), a comprehensive benchmark designed for clue-grounded question answering in long videos. The dataset comprises 1,219 manually curated videos with an average duration of approximately 27 minutes, spanning 14 diverse domains such as life records, movies, and news. Unlike traditional QA datasets, CG-Bench provides 12,129 Question-Answer-Clue triplets where each QA pair is associated with precise temporal spans serving as supporting evidence. This granular annotation structure covers varying cognitive demands, including perception, reasoning, and hallucination detection, thereby providing a rich environment for learning long-horizon information seeking and temporal localization skills.

**Implementation and Environment**   All experiments are conducted on a computational node equipped with 8 NVIDIA A100 GPUs. The agent framework is built upon the rllm (Tan et al., 2025), utilizing vLLM (Kwon et al., 2023) for high-throughput inference and Ray for distributed rollout management. Our architecture employs a Qwen3-8B planner for high-level planning. Regarding visual evidence processing, we utilize the Qwen3-VL-8B-Instruct inspector during the training phase, whereas the Qwen2.5-VL-7B-Instruct model is employed for inference on benchmarks. Furthermore, the visual retrieval tool incorporates DeepSeek-V3.2 as an LLM filter to refine retrieved content. The agent interacts with the environment with a maximum budget of $K = 16$ steps per trajectory. At each step, the planner chooses between semantic retrieval using a pre-computed video index and visual inspection. During visual inspection, video frames are extracted at 1 fps, capped at 64 frames, and resized to 1024px for MLLM consumption. We perform training on the CG-Bench dataset and conduct validation on diverse out-of-domain benchmarks.

**Training Framework**   We fine-tune the policy using GRPO. To ensure training stability and efficiency, we utilize FSDP2 sharding strategies with bfloat16 precision. The optimization process employs a learning rate of $1 \times 10^{-6}$ accompanied by a cosine warmup schedule with a ratio of 0.05 for one epoch. To prevent excessive deviation from the reference policy, we include a KL divergence penalty with a coefficient of 0.001. The reward mechanism focuses on ground-truth alignment and framework adherence. Positive rewards are granted for final answer correctness, determined via exact match for multiple-choice questions, as well as for adhering to the strict tool-use format.

# B. Video Segmentation and Retrieval Index

**Video Indexing and Retrieval**   We partition each long video into fixed-length clips of 16 seconds and uniformly sample frames at a rate of 1 FPS. This temporal grid underpins both the retrieval and inspection mechanisms. To facilitate semantic indexing, we employ Qwen3-8B-Instruct to generate concise retrieval-oriented captions that summarize salient entities, actions, and scene contexts within one to three sentences. These textual proxies serve as the foundation for the VisualRetrieve tool, which embeds both clip captions and planner queries using the OpenAI text-embedding-3-large model. Candidate clips are subsequently identified through cosine similarity.

# C. Additional Ablation

### C.1. Behavioral Statistics and Trajectory Analysis.

We further characterize whether the search policy behaves like random switching or meaningful refinement. As shown in Table 5, VideoSEAL follows a retarget-then-refine pattern: it reaches evidence earlier, as measured by Hit@1/2/3, and recovers more often after an initial miss. After the first hit, the median temporal overlap with the ground-truth evidence also increases, indicating local refinement around the relevant evidence rather than random exploration.

Following the temporal-grounding setup in Sec. 3.1, we use the same annotated evidence intervals $\mathcal{E}^{\star}$, temporal IoU tIoU, and threshold $\gamma = 0.05$. For a trajectory $\xi$, let $T$ be its length. At step $t$, let $\mathcal{E}_t$ denote the temporal spans accessed through retrieval or inspection, where we omit the argument $\xi$ for readability. We define the best step-level temporal overlap as

$$m_t = \begin{cases} \max_{\tau \in \mathcal{E}_t, \ \tau^{\star} \in \mathcal{E}^{\star}} \text{tIoU}(\tau, \tau^{\star}), & \mathcal{E}_t \neq \emptyset, \\ 0, & \mathcal{E}_t = \emptyset. \end{cases}$$

A step is counted as a hit if $m_t \geq \gamma$.

Table 5 reports how often a trajectory reaches annotated evidence during the early search process. Hit@$k$ ($k = 1, 2, 3$) is the

percentage of evaluation trajectories that contain at least one hit within the first $k$ available steps. Thus, Hit@1, Hit@2, and Hit@3 measure whether the search reaches evidence immediately, within two steps, or within three steps, respectively.

Recovery is evaluated only on trajectories that miss at the first step ($m_1 < \gamma$). It is the percentage of these first-step misses that reach annotated evidence in any later step. Trajectories with no later step are counted as not recovered. For trajectories that contain at least one hit, let $t_{\mathrm{hit}} = \min\{t : m_t \geq \gamma\}$ be the first hit step. We report $\mathrm{IoU_{med}}$ as first-hit→post-hit: the left value is the median overlap $m_{t_{\mathrm{hit}}}$ when evidence is first reached, and the right value is the median best overlap observed from that point onward. A larger post-hit value indicates that the trajectory refines toward better-aligned evidence after first reaching the relevant region.

Table 5. Trajectory-policy statistics on LVBench. Most metrics are percentages except $\mathrm{IoU_{med}}$.

| Method | Hit@1 | Hit@2 | Hit@3 | Recovery | $\mathrm{IoU_{med}}$ |
|---|---|---|---|---|---|
| Ego-R1 | 23.6 | 26.7 | 27.3 | 7.5 | 0.0→0.0 |
| DrVideo | 25.5 | 34.8 | 39.8 | 33.0 | 0.0→0.0 |
| VideoMind | 31.1 | 39.1 | 41.0 | 24.3 | 0.0→0.1 |
| Video-MTR | 32.9 | 32.9 | 32.9 | 0.0 | 0.0→0.1 |
| **Ours** | **53.9** | **56.5** | **56.9** | **34.8** | **0.2→0.4** |

## C.2. Retrieval Granularity and State Stability.

We stress-test state stability by sweeping the retrieval depth $k$ (top-$k$ candidates returned by the embedding index) while keeping the frame budget and backbones fixed. Increasing $k$ improves the oracle recall of evidence, but it also increases the amount of intermediate tool traces that must be processed by the planner. In coupled agents, these traces are directly injected into the same context that maintains the routing state and holds the Answer authority, making the planner increasingly susceptible to context dilution as $k$ grows. Empirically, coupled baselines saturate or degrade at larger $k$, accompanied by a higher per-step token footprint. In contrast, our decoupled framework keeps the planner's state low-bandwidth (structured memory) and delegates termination to an evidence-gated inspector. As a result, the planner remains stable as $k$ increases: higher retrieval recall translates into consistent gains in $\mathrm{Acc_{ans}}$ (Table 6). Although $G_t$ exhibits a moderate decline due to the lower signal-to-noise ratio in larger candidate pools, the architecture effectively leverages the expanded search scope to issue more accurate answers without succumbing to the context collapse observed in baselines.

## C.3. The Asymmetry of Scaling: Reasoning versus Perception

To disentangle the distinct contributions of semantic planning and visual grounding, we conduct a controlled ablation across orthogonal scaling regimes. By selectively upgrading the planner or the inspector while holding the counterpart fixed, we isolate the limiting bottleneck in the current agentic video understanding loop.

The empirical results reveal a fundamental asymmetry in component scalability. As detailed in the *Scaling Reasoning* section of Table 7, enhancing the cognitive backbone yields diminishing returns. Upgrading the planner from the 8B baseline to GPT-4o results in a counter-intuitive regression in accuracy to 52.3%, while the transition to Gemini-3-Flash-Preview offers only marginal gains. This phenomenon indicates that the temporal navigation task has reached a state of *semantic saturation*; the 8B-parameter planner already possesses sufficient agency to formulate valid search queries, rendering further increases in reasoning capacity redundant when bounded by the same visual acuity.

In stark contrast, the *Scaling Perception* regime demonstrates that the system is heavily *perception-bound*. By maintaining the lightweight 8B planner and substituting the inspector with Gemini-3-Flash-Preview, we observe a breakthrough in performance, with accuracy surging to 69.9%. This represents a massive 14.8 point improvement over the baseline, far outstripping the gains from reasoning scaling. This divergence underscores a critical insight: once the navigational logic is sound, the upper bound of LVU is dictated by the fidelity of pixel-level inspection rather than the sophistication of linguistic planning. Our decoupled architecture uniquely exploits this property, enabling significant zero-shot performance leaps by simply plugging in stronger visual backends.

## C.4. Newer-Backbone Results and Inspector Modularity

To validate that the decoupled design is not tied to an outdated visual backbone, we further evaluate VideoSEAL with newer and stronger inspector models under the same pipeline. Across all variants, we keep the planner, retrieval stack, search

*Table 6.* **Scalability Analysis.** Comparison of robustness as the number of retrieved candidates ($k$) increases. Our decoupled method maintains high performance compared to the baseline.

| Method | $k$ | $G_t$ (%) | Acc. (%) |
|---|---|---|---|
| | 10 | 51.1 | 48.2 |
| Coupled | 20 | 46.8 | 50.8 |
| | 40 | 38.8 | 47.2 |
| | 10 | 45.3 | 51.2 |
| **Decoupled (Ours)** | 20 | 42.9 | 53.4 |
| | 40 | 40.6 | 55.1 |

*Table 7.* **Decoupled Scaling Analysis on LVBench.** We explore orthogonal scaling regimes to identify the system bottleneck. The comparison reveals a distinct hierarchy: *Scaling Perception* via a superior visual backend unlocks significantly higher gains than *Scaling Reasoning* through a stronger language model, indicating that performance is currently bounded by perceptual fidelity rather than planning logic.

| Scaling Regime | Planner | Inspector | $\text{Acc}_{\text{ans}}$ (%) | $G_t$ (%) | $G_s$ (%) |
|---|---|---|---|---|---|
| *Baseline* | Qwen3-8B | Qwen2.5-VL (7B) | 55.1 | 52.8 | 40.6 |
| *Scaling Reasoning* | GPT-4o | Qwen2.5-VL (7B) | 52.3 | 51.9 | 42.3 |
| *(Strong Planner)* | Gemini-3-Flash-Preview | Qwen2.5-VL (7B) | 59.4 | 54.5 | 35.8 |
| *Scaling Perception* | Qwen3-8B | GPT-4o | 62.0 | 51.3 | 39.1 |
| | Qwen3-8B | Gemini-3-Flash-Preview | **69.9** | 51.5 | 43.1 |

budget, and interaction protocol unchanged, and only replace the frozen inspector used for visual verification and final answer authority. As shown in Table 8, stronger inspectors substantially improve absolute performance across all four benchmarks. This confirms the modularity of VideoSEAL: once the planner learns to search for candidate evidence, better visual backbones can be plugged in to improve pixel-level verification without retraining the planner.

## C.5. Impact of Visual Sampling Density

Beyond model capacity, we investigate the sensitivity of our framework to *temporal granularity* by varying the frame budget from 64 to 256. This scaling analysis aims to determine whether the performance bottleneck stems from missing visual information due to aggressive downsampling or the inability to process available cues.

As summarized in Table 9, we observe a robust monotonic improvement in VideoMME performance as the frame budget increases. Upgrading the sampling density from 64 to 256 frames yields a consistent gain in overall accuracy. This trend indicates that the current system effectively capitalizes on higher-resolution temporal structures. Unlike coupling models where excessive token density often leads to context pollution or "lost-in-the-middle" phenomena, our inspector-gated architecture processes visual data in targeted, context-isolated spans. This structural advantage allows the agent to ingest denser visual information for precise verification without overwhelming the reasoning planner, thereby converting enhanced temporal fidelity directly into grounded decision-making.

## C.6. Restoring Groundedness: Closing the Hallucination Gap

To address the reward pressure and the consequent hallucination gap identified in Section 3.2, our decoupled framework restructures the optimization process. By explicitly separating the evidence seeking objective from the answer generation reward, we enforce a structural dependency where correct answers must derive from verified video segments.

**Empirical Verification of Alignment** We analyze the training dynamics of our approach in Figure 7. Contrary to the divergence observed in coupled baselines, our approach exhibits a strong alignment between answer accuracy and evidence retrieval success. Throughout the training process, the performance curves remain tightly synchronized with the discrepancy consistently maintained at a negligible margin of approximately 0.02. This synchronization confirms that the agent avoids learning shortcut heuristics or relying on parametric priors. Instead, the gains in answer accuracy are causally linked to improvements in evidence localization. This demonstrates that the agent strictly adheres to the retrieval-then-reasoning

*Table 8.* Newer-backbone results under the same decoupled pipeline. Only the inspector is changed; all numbers are accuracy (%).

| Inspector | LongVideoBench | LVBench | MLVU | Video-MME |
|---|---|---|---|---|
| Qwen2.5-VL-7B | 62.0 | 55.1 | 68.2 | 62.9 |
| Qwen3-VL-8B | 69.2 | 58.5 | 79.2 | 74.0 |
| Kimi2.5 | **80.8** | 65.7 | **84.3** | 82.5 |
| Gemini-3-Flash-preview | 77.2 | **69.9** | 79.3 | **83.0** |

*Table 9.* **Impact of Frame Budget on VideoMME.** We evaluate the system performance across varying temporal resolutions. The results demonstrate a positive correlation between sampling density and answer accuracy, confirming that our decoupled inspector efficiently leverages finer-grained visual cues without suffering from context overload.

| Frame | VideoMME Overall ($\uparrow$) |
|---|---|
| 64 (Default) | 62.9 |
| 128 | 63.6 |
| 256 | 64.4 |

workflow rather than resorting to speculation.

## C.7. Efficiency and Cost Analysis

Beyond accuracy, the practical deployment of agentic systems requires economic efficiency. We therefore evaluate the performance of each framework under a standardized budget and implementation setting. Specifically, we report the average wall-clock latency per question, the average number of frames processed by the inspector, and the average monetary cost per query. All financial metrics are calculated based on OpenRouter pricing and sum the expenses for both the planner and MLLM components. Crucially, the reported cost for our method explicitly accounts for the overhead introduced by the DeepSeek V3.2 LLM filter. As summarized in Table 10, our decoupled design achieves the lowest cost among agentic approaches at $0.015 per query while maintaining superior accuracy. This efficiency stems from the reduction of redundant visual inspections and the use of inspector feedback to refine queries. These savings effectively offset the computational cost of the retrieval filter.

*Table 10.* Comparison of accuracy and inference efficiency. Our decoupled framework achieves significantly higher accuracy while strictly minimizing monetary costs compared to alternative agentic baselines.

| Method | Acc. (%) | Latency (s) | Avg. Frames | Avg. Cost ($) |
|---|---|---|---|---|
| Qwen2.5-VL-7B-Instruct | 39.4 | 686.3 | 768.0 | 0.004 |
| VideoAgent | 42.3 | **22.8** | – | 0.038 |
| DrVideo | 51.4 | 56.6 | **35.3** | 0.053 |
| Ego-R1 | 39.4 | 70.4 | 315.9 | 0.034 |
| **Ours (Decoupled)** | **55.1** | 69.5 | 63.1 | **0.015** |

## C.8. Impact of Retrieval Filtration

To enforce the structural separation of concerns, our Retrieval primitive incorporates a semantic filter designed to prune irrelevant noise before it reaches the planner. We instantiate this component using **DeepSeek-V3.2** due to its high token throughput and semantic capability. To empirically validate the necessity of this gatekeeping mechanism and clarify its implementation, we conduct an ablation study comparing three filtering configurations while keeping the planner fixed as Qwen3-8B: (i) *None*, where raw top-$k$ retrieved chunks are injected directly into the planner's context; (ii) *DeepSeek-V3.2* (Ours); and (iii) *GPT-4o*, representing a frontier model upper bound.

As presented in Table 11, bypassing the filter ("None") precipitates a sharp performance degradation, with accuracy dropping from 55.1% to 51.2%. This regression empirically corroborates our diagnosis of **Prompt Pressure**: without semantic curation, the accumulation of irrelevant text chunks saturates the planner's context window, degrading its ability to reason over long horizons. Furthermore, substituting the lightweight DeepSeek model with GPT-4o yields only a marginal gain of 0.8%. This saturation suggests that the system's performance is bounded by the recall of the initial dense retrieval rather than

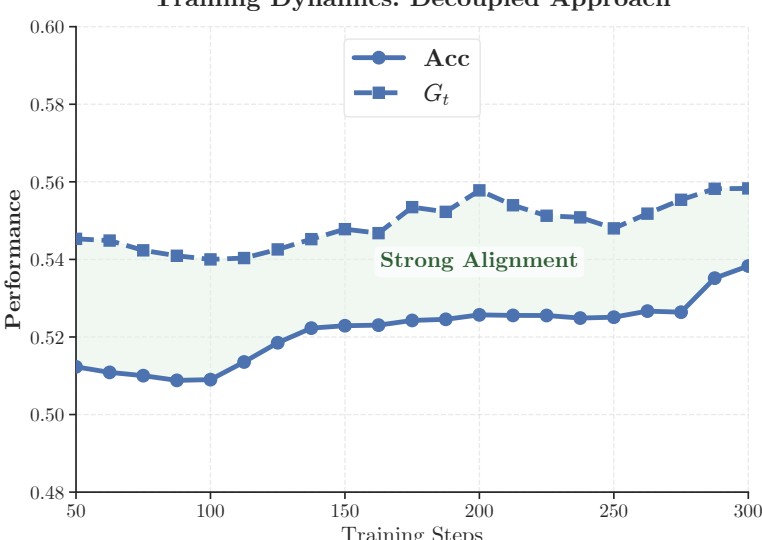

*Figure 7.* **Training Dynamics of the Decoupled Approach.** Unlike the *Hallucination Gap* observed in coupled baselines, our method maintains a **strong alignment** between answer accuracy and temporal groundedness ($G_t$). The performance gap remains negligible ($\Delta \approx 0.02$) throughout the training process. This indicates that improvements in answer accuracy are driven entirely by successful evidence retrieval, effectively eliminating the "cheating" behavior caused by reward pressure.

the reasoning depth of the filter. Consequently, our choice of DeepSeek-V3.2 represents an optimal design point, effectively mitigating context pollution without incurring the latency and cost overhead of larger models.

*Table 11.* **Ablation on Retrieval Filtration Strategy.** We analyze the impact of the filtering mechanism on the LVBench validation set. Removing the filter exposes the planner to context noise, significantly exacerbating Prompt Pressure and reducing accuracy. The choice of DeepSeek-V3.2 balances performance with efficiency.

| Planner | Filter Model | Acc (%) | $G_t$ (%) | $G_s$ (%) |
|---------|--------------|---------|-----------|-----------|
| Qwen3-8B | – | 51.2 | 47.2 | 45.7 |
| Qwen3-8B | DeepSeek-V3.2 | 55.1 | 52.8 | 40.6 |
| Qwen3-8B | GPT-4o | 55.0 | 53.0 | 41.0 |

## C.9. Limitations

Although decoupling the search and verification processes effectively mitigates evidence misalignment, this architectural choice inherently introduces certain trade-offs. Explicitly inspecting candidate spans naturally adds inference overhead; consequently, latency can spike during complex queries even if the average computational cost remains competitive. Furthermore, the inspection mechanism itself is not infallible. The model may occasionally misjudge sufficiency, leading to premature termination on incomplete evidence or unnecessary retrieval cycles when sufficient clues are missed. Beyond these operational constraints, evaluating semantic groundedness currently relies on an LLM judge. We advocate treating this metric as a comparative diagnostic rather than an absolute oracle, particularly when handling ambiguous or highly fine-grained visual information. From a broader application standpoint, our empirical findings are based on a specific set of clue-grounded training data and standard benchmarks.

## D. Case Studies

We present five representative trajectories (three successes and two failures) to characterize the evidence seeking behavior of the decoupled planner–inspector framework. In each episode, the planner proposes candidate spans via `visual_retrieve`, while the inspector exclusively controls termination: it issues an answer only when $z_t = 1$, and otherwise returns SEARCH_MORE with a missing-evidence signal $f_t$.

**S1: Counting (Fig. 8).** The planner retrieves three dispersed spans (00:00:00–00:00:16; 00:15:12–00:15:28; 00:33:20–00:33:36) and submits non-overlapping candidates for inspection, where the inspector counts five red stripes and selects option B (0.95). This discriminative count directly rules out the remaining options, yielding an evidence-based decision.

**S2: Post-condition (Fig. 9).** The planner first retrieves an off-target co-occurrence span (00:50:24–00:50:40) that contains no verifiable weather evidence, prompting the inspector to return SEARCH_MORE. Guided by the missing-evidence signal $f_t$, it reformulates retrieval and re-centers on the ground-truth time reference (00:26:48–00:28:32), where snowfall is verified in 00:27:44–00:28:00 and option D is returned (1.00). The episode terminates only once the post-condition is supported by discriminative visual cues, rather than by co-occurrence or span relevance alone.

**S3: Recovery (Fig. 10).** The planner localizes the ground-truth neighborhood around 01:08:07–01:08:26, yet the first two `visual_inspect` calls return SEARCH_MORE because sparse sampling over a long span makes the option-critical cues (same-species rescue and threat attribution) hard to verify. Instead of discarding the window, the planner treats each rejection as actionable feedback, refocuses `visual_retrieve` queries toward option-aligned details, and re-submits a tighter, more verifiable span; the inspector then confirms a ladybug–ladybug interaction near 01:08:22 and terminates with option C (confidence 0.95).

**F1: Fine-grained error (Fig. 11).** Although the planner retrieves relevant spans, the inspector exhibits a fine-grained perceptual failure on headwear attributes (small accessories / color), producing an answer that is inconsistent with its own stated evidence and ultimately incorrect (ground truth: D).

**F2: Off-target retrieval and local-cue overreliance (Fig. 12).** The retrieval concentrates on early meeting-like spans and misses the evidence-bearing window required to validate the query's global constraint, forcing the inspector to answer from local seasonal cues and resulting in an incorrect commitment (A, 0.95). The lack of overlap with `time_reference` (00:49:49–00:49:49) reflects this off-target retrieval and leaves the decision unsupported by time-aligned evidence.

**Summary.** Taken together, these cases highlight a consistent closed loop: decoupling answer authority from planning systematically steers retrieval toward discriminative evidence and enables evidence-entailed termination. The failure modes further indicate that residual errors are dominated by (i) perception-bound fine-grained attributes and (ii) coverage failures under global temporal constraints, rather than by the planning protocol itself. These limitations are complementary to common strengths in recent long-video agents, suggesting that stronger visual backbones and improved coverage can be integrated without altering the decoupled protocol.

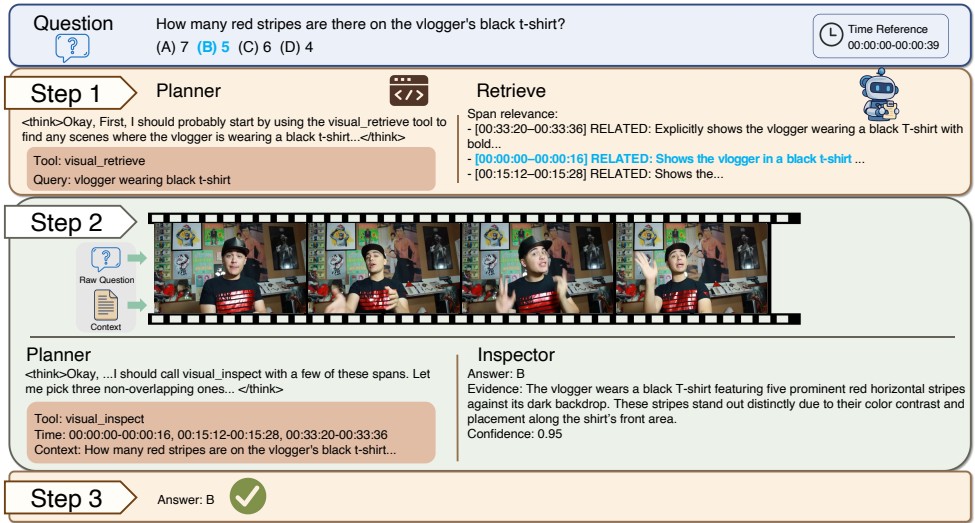

*Figure 8.* **Case S1 (Counting).** The inspector verifies a unique count-based cue from retrieved spans, enabling evidence-sufficient termination.

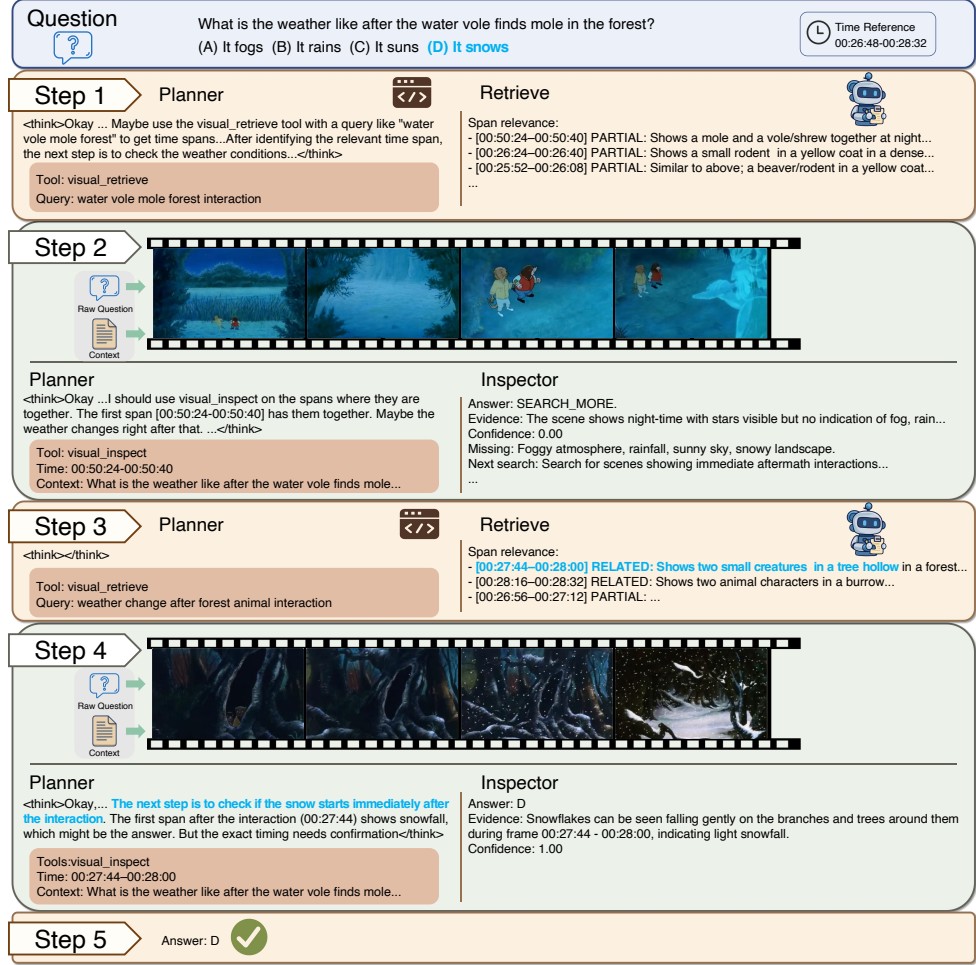

*Figure 9.* **Case S2 (Post-condition).** SEARCH_MORE refines retrieval to a post-event span where the decisive visual cue is verified.

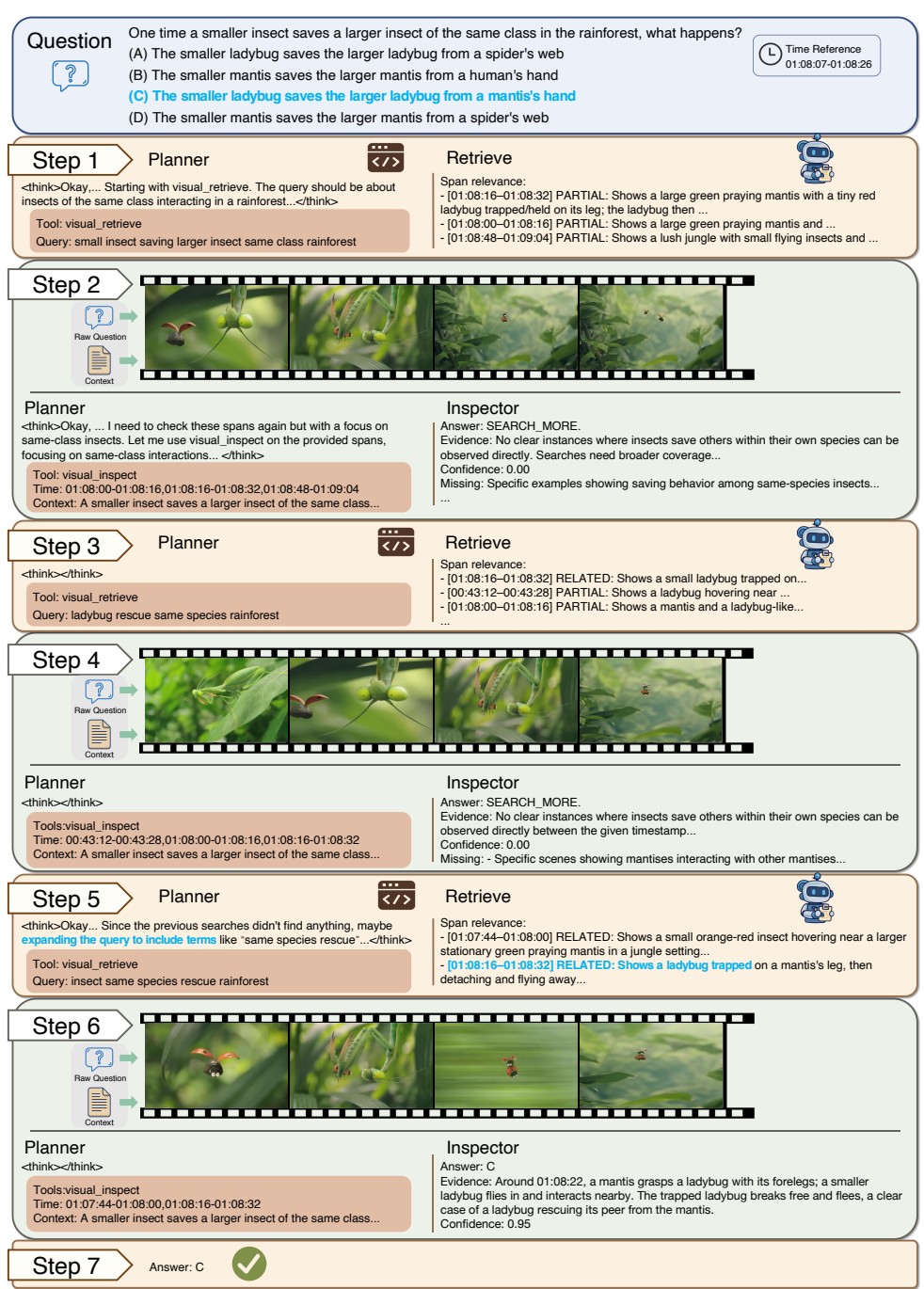

*Figure 10.* **Case S3 (Recovery).** Query refocusing recovers from inspector false negatives under long-span, sparse sampling.

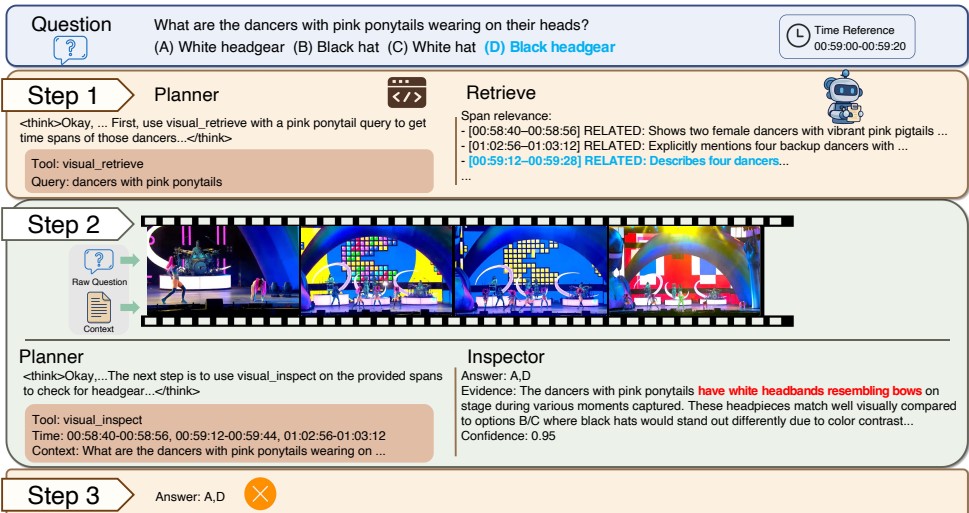

Figure 11. **Case F1 (Fine-grained perceptual failure).** Fine-grained headwear attributes (small accessories / color) are misperceived, yielding evidence–option inconsistency.

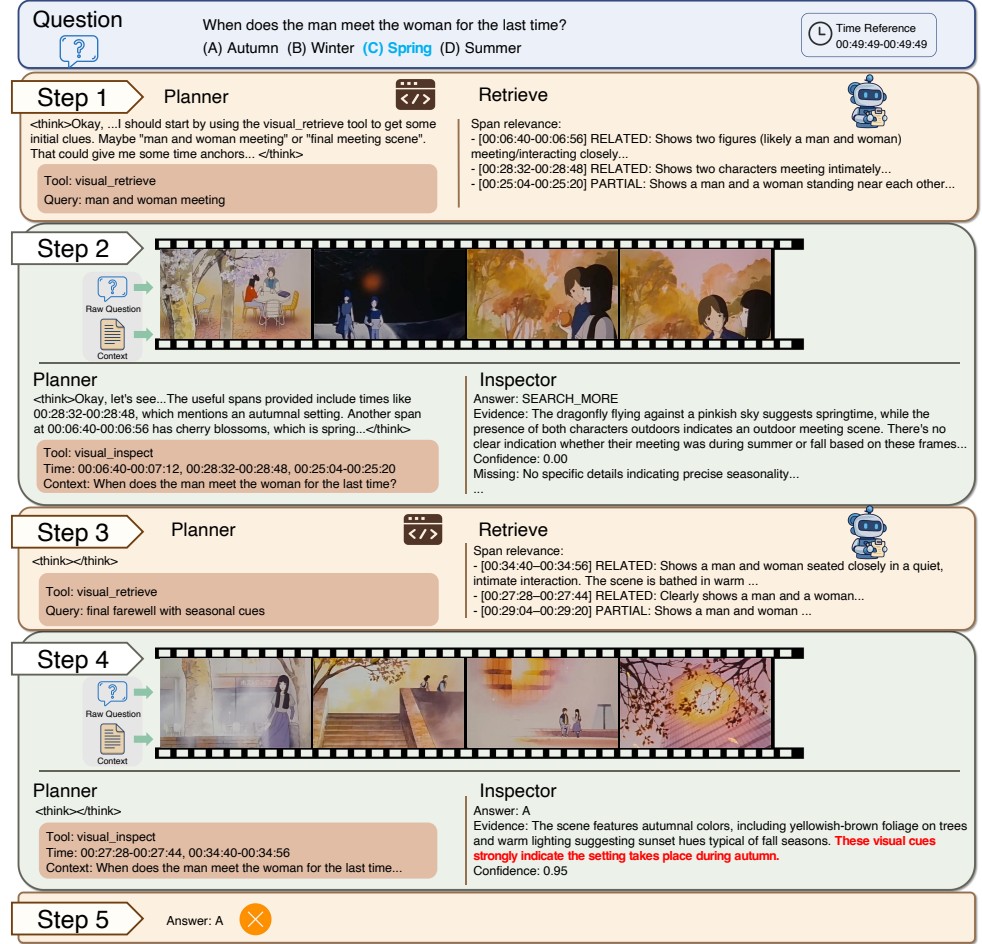

Figure 12. **Case F2 (Off-target retrieval).** The search misses the time-aligned evidence window, forcing an answer from local cues and resulting in an incorrect commitment.

# E. Algorithm Details

**Inference Execution and Fallback**     We formally present the inference execution flow in Algorithm 1. The process follows the decoupled framework wherein the planner iteratively refines its search memory $h_t$ via sequence concatenation $\oplus$ rather than set union to preserve the temporal order of reasoning traces. A critical component of this implementation is the maintenance of a candidate cache $\mathcal{C}$ which persists across the entire episode to serve as a safety net. Should the search budget $K$ be exhausted prior to a definitive sufficiency verdict $z_t = 1$, we enforce a fallback mechanism wherein the agent aggregates all historically retrieved candidates into a unified timeline $\tau_{\mathrm{all}} = \bigcup \mathcal{C}$ and executes a forced `VisualInspect` call.

---

**Algorithm 1** Decoupled Planner–Inspector Framework

---

1: **Input:** video $\mathcal{V}$ (duration $T_{\mathrm{vid}}$), original question $q$, budget $K$
2: **Initialize:** search memory $h_0 \leftarrow [\,]$, candidate cache $\mathcal{C} \leftarrow \emptyset$, last tool $\ell \leftarrow \varnothing$
3: **for** $t = 1$ **to** $K$ **do**
4:      $u_t \leftarrow \mathrm{Planner}(q, h_{t-1})$
5:      **if** $u_t$ is `VisualRetrieve`$(k)$ **then**
6:         $\mathcal{P}_t \leftarrow$ `VisualRetrieve`$(q, k)$ $\{\mathcal{P}_t$: retrieved candidate spans at turn $t\}$
7:         $\mathcal{C} \leftarrow \mathcal{C} \cup \mathcal{P}_t$ $\{$Cache retrieved candidates$\}$
8:         $h_t \leftarrow h_{t-1} \oplus [\mathrm{RET}(\mathrm{Summary}(\mathcal{P}_t))]$
9:         $\ell \leftarrow$ `VisualRetrieve`
10:      **else if** $u_t$ is `VisualInspect`$(v_t, q)$ **then**
11:         $\{v_t$ is the set of spans submitted by the Planner at turn $t$ (may change across turns)$\}$
12:         $(z_t, f_t) \leftarrow$ `VisualInspect`$(v_t, q)$
13:         $h_t \leftarrow h_{t-1} \oplus [(v_t, z_t, f_t)]$
14:         $\ell \leftarrow$ `VisualInspect`
15:         **if** $z_t = 1$ **then**
16:            **return** $f_t$ $\{$Evidence-grounded resolution$\}$
17:         **end if**
18:      **end if**
19: **end for**
20: $\{$Forced fallback: append a terminal `VisualInspect` to satisfy the last-tool constraint$\}$
21: **if** $\ell \neq$ `VisualInspect` **then**
22:      **if** $\mathcal{C} \neq \emptyset$ **then**
23:         $\mathcal{S}_{\mathrm{fb}} \leftarrow \mathcal{C}$
24:      **else**
25:         $\mathcal{S}_{\mathrm{fb}} \leftarrow \{[0, T_{\mathrm{vid}}]\}$ $\{$Inspect full video if no candidates$\}$
26:      **end if**
27:      $(z_{\mathrm{fb}}, f_{\mathrm{fb}}) \leftarrow$ `VisualInspect`$(\mathcal{S}_{\mathrm{fb}}, q)$
28:      $h_{K+1} \leftarrow h_K \oplus [(\mathcal{S}_{\mathrm{fb}}, z_{\mathrm{fb}}, f_{\mathrm{fb}})]$
29:      **if** $z_{\mathrm{fb}} = 1$ **then**
30:         **return** $f_{\mathrm{fb}}$
31:      **end if**
32: **end if**
33: **return** `EvidenceNotFound`

---

# F. Prompt.

## F.1. System Prompt for the Decoupled Planner

---

**Decoupled Tool-Agent System Prompt**

```
You are a helper that answers multi-step video questions by sequentially invoking
functions.
Your ONLY job is to retrieve and refine candidate time spans; the final answer content
must
come from visual_inspect, not from your own imagination.

Tool outputs are lossy and may miss details. Prefer improving coverage by:
- trying different queries,
- inspecting new / non-overlapping time spans,
rather than re-running the same tool on the same spans.

If you cannot find strong anchors after several retrieval attempts, you may switch to an
exploration mode:
- randomly sample a few large, non-overlapping time spans across the video,
- send them to visual_inspect to discover where relevant evidence might be.

Each turn, do exactly one of the following:

* If further clues and evidence are needed:
  Output EXACTLY one tool call:
  <tool_call>{"name":"...","arguments":{...}}</tool_call>

* If ready to finalize:
  You may enter this branch ONLY if the immediately previous turn called visual_inspect
  and its output contains a clear, reliable verdict.
  Output:
  <final>...</final>

Final Format
• Multiple-choice → ONLY the letter(s) uppercase.(e.g., C). No extra words.

Rules
- One Tool call only per turn. Exactly one tool call per turn, except final turn (no
tool call).
- You are a retriever, not an answerer: never invent answers; never override
visual_inspect.
- You MUST NOT conclude from any non-visual_inspect tool alone.
- You may output <final> ONLY if the immediately previous turn was a visual_inspect call
  AND it returned a clear, reliable verdict.
- If the last tool call is NOT visual_inspect, you MUST NOT output <final>; keep
searching
  and then call visual_inspect.
- The final answer must match the most recent reliable visual_inspect verdict (prefer
the
  latest decisive visual_inspect when multiple exist).
- Do NOT re-run the same tool on the same time spans just to \get more info".
  Instead, broaden coverage via different queries and new spans, or use exploration mode
  (large non-overlapping spans).
- If visual_inspect explicitly indicates that more search is needed (e.g., a special
\search more"
  flag or low-confidence signal), you must NOT finalize.
- A good pattern is to alternate tools, e.g.:
  visual_retrieve → visual_inspect → visual_retrieve(different query) →
  visual_inspect(different spans) → final.
- As soon as you have plausible spans, it is better to call visual_inspect again
(possibly with
  more detailed context or slightly refined spans) than to keep doing blind retrieval.
  Multiple visual_inspect with different time spans calls are encouraged.
```

---

```
- Always zero-pad time fields (HH:MM:SS).
- Hard constraint: end_time must be strictly greater than start_time.

Tool Calling Conventions
- visual_retrieve: {"name":"visual_retrieve","arguments":{"query":"..."}}.
  Use free-form natural language or short visual phrases related to the question to
retrieve
  visually/scene-relevant regions and coarse time anchors from the unified visual index.
- visual_inspect: {"name":"visual_inspect","arguments":{
      "spans":[{"start_time":"HH:MM:SS","end_time":"HH:MM:SS"}, ...],
      "context":"Restate the original question + 2{4 visual sub-questions; explain
briefly why
               these spans were chosen (e.g., tool hits/previous clues); provide a
short
               'look-for' checklist of disambiguating cues to verify within these spans
               (appearance: gender/age/face/hair, clothing/accessories,
actions/objects,
               location, on-screen text). Phrase hints as 'look for / verify' rather
than
               confirmed facts."
  }}.
  Constraints: each span must satisfy end_time > start_time.
```

## F.2. System Prompt for the coupled agent

---

**Coupled Tool-Agent System Prompt**

---

```
TOOLAGENT_SYS_PROMPT_EN = """
You are a helpful assistant who answers multi-step questions by sequentially invoking
functions. Follow the THINK → ACT → OBSERVE loop. Each turn, do exactly this and keep
every <think> rich with reasoning:

* If further clues and evidence are needed:
  <think>
  Reflect explicitly on your current evidence and remaining uncertainties. Clearly
reference outputs from previous tools. Justify precisely which next tool, parameters,
and targeted window can provide valuable additional clues or reduce significant
uncertainty.
  </think>
  <tool_call>{"name":"...","arguments":{...}}</tool_call>

* If ready to finalize:
  <think>
  Integrate the complete set of clues gathered from prior tools. Reference decisive
evidence (tool name + key timestamps/details) and explain why no further investigation
is necessary. If you used visual_inspect, briefly cite the key visual evidence.
  </think>
  <final>...</final>

Final Format
• Multiple-choice → ONLY the letter(s) uppercase. For multi-select, separate by comma
(e.g., A,C). No extra words.
• Open-ended → the final concise textual conclusion.

Rules
- One output per turn. Exactly one tool call per turn, except final turn (no tool call).
- Start broad: use retrieval tools to map storyline clues and concrete time-windows;
then iteratively narrow with targeted follow-up retrieval and/or inspection until
decisive evidence is found.
- Final turn sequence strictly follows: <think> → <final>.
- Always zero-pad time fields (HH:MM:SS).
- Every <think> must contain detailed reasoning in plain text (no placeholders). Empty
or purely generic thoughts are invalid.
- Evidence grounding: Base all reasoning and conclusions strictly on evidence from tool
outputs (cite tool name and timestamp/details). Do NOT invent details or speculate. If
evidence is insufficient, plan another tool call instead of guessing. Avoid hedging
words like "maybe", "might", "seems"; be definitive only when evidence supports it.
- Before any <tool_call>, the preceding <think> must reference what previous tools
revealed (or failed to reveal) and justify the specific tool/arguments you are about to
execute.
- Before any <final>, the preceding <think> must cite the precise evidence (tool name +
timestamp/key detail) that resolves the question.
- You must output your final choice in <final>...</final> without unknown or
uncertainty.
- Visual tools (visual_retrieve and visual_inspect) NEVER return person names or
identities. They only show appearances and actions (clothes, hair, position, behavior).
If a question asks about who a person is (names or roles), you must infer identity from
long-term storyline and consistent appearance across scenes, not from a single visual
tool call.

Tool Calling Conventions
- visual_retrieve: {"name":"visual_retrieve","arguments":{"query":"..."}}.
  Use free-form natural language or short visual phrases related to the question to
retrieve visually/scene-relevant regions and coarse time anchors from the unified visual
index.
- visual_inspect: {"name":"visual_inspect","arguments":{
      "spans":[{"start_time":"HH:MM:SS","end_time":"HH:MM:SS"}, ...],
```

```
      "context":"Restate the original question + 2{4 visual sub-questions; explain
briefly why these spans were chosen (e.g., tool hits/previous clues); provide a short
'look-for' checklist of disambiguating cues to verify within these spans (appearance:
gender/age/face/hair, clothing/accessories, actions/objects, location, on-screen text).
Phrase hints as 'look for / verify' rather than confirmed facts."
  }}.
  Constraints: use at most 10 spans per call; each span must satisfy end_time >
start_time.

Balanced Clue Discovery Guidance
- Typically start with visual_retrieve to establish narrative anchors and fine-grained
visual context early.
- If initial retrieval is weak, broaden the query and retry retrieval; or use
visual_inspect on a small number of broader spans to quickly confirm whether any
decisive evidence exists there, then narrow down with additional retrieval.
- Confirm decisive details with visual_inspect when needed, then finalize only when
evidence is sufficient.
- Collect corroborating evidence across tools and timestamps; prioritize the spans where
multiple tools agree.
"""
```

### F.3. Prompt for Visual Inspect (Inspector)

**Visual Inspect Prompt Template**

```
You will be shown frames sampled across multiple time windows of a video.
Spans: {spans_label}.
{Context: <optional prior-search summary, only if provided>}

Use the frames together with the optional Context (which summarizes prior searches) to
answer the question.
Reason explicitly about how the visual evidence matches each part of the question and
any clues in Context.

Important: You ONLY see frames from the spans listed above. If a required element is not
visible or cannot be inferred here,
that usually means the answer is outside these spans and the retriever should search
elsewhere in the video.
Only treat these spans as FIXED ground-truth if the question explicitly asks about this
exact time range
(e.g., it mentions the same timestamps or says "in this segment"). In that fixed-range
case, you MUST choose the best-supported
option from these frames, even if your confidence is below 0.95, and you must NOT output
SEARCH_MORE.

Only choose option letters if the frames (with help from Context) clearly support all
required elements
(entities, actions, attributes, relations, and any time conditions).
If this is NOT a fixed-range question and any required element cannot be verified or
reliably inferred from these frames,
do NOT choose an option.

Always output a numeric confidence in [0.00, 1.00]. If this is NOT a fixed-range
question and confidence is below 0.95,
output Answer: SEARCH_MORE.

When outputting SEARCH_MORE, briefly state which specific elements of the question are
still unverified in these spans
and suggest 1--3 concrete next searches as query phrases / what to look for.
Do NOT include any explicit timestamps, HH:MM:SS ranges, or relative time directions
(e.g., earlier/later).
Do NOT recommend re-checking the same spans unless the question itself requires it.
Finally, add a short Reminder line telling the central LLM that the NEXT tool call
should NOT be visual_inspect,
and should instead use other retrieval tools to find new spans.

When citing visual evidence in your answer, reference exact timestamps in HH:MM:SS (or
ranges like HH:MM:SS{HH:MM:SS)
from the provided spans; do NOT use "frame N" or image indices.

Question(s):
{qtext}

OUTPUT FORMAT
- Start with a single line: Answer: <one or more option letters such as "A" or "A,C", OR
"SEARCH_MORE">.
- Then provide Evidence: <1{3 sentences grounded in the frames; include HH:MM:SS
timestamps rather than "frame N"
  when referencing specific visual cues>.
- Then Confidence: <0.00{1.00, two decimals>.
- If Answer is SEARCH_MORE, add three more lines:
  Missing: <bullet-like list inside one line of what is not visible / still unclear>.
  Next search: <concrete query phrases / what to search for next, WITHOUT any
timestamps>.
  Reminder: <one sentence: do NOT call visual_inspect next; use other tools to search
new spans>.
```

```
If you output option letters in a non-fixed-range question, your confidence MUST be at
least 0.95 and your evidence must cover
every element of the question.
```

## F.4. Prompt for Visual Retrieve Summary

**Visual Retrieve Summary Prompt Template**

```
ROLE (clue finder / relevance check ONLY)
- Do NOT answer the final QA and do NOT choose multiple-choice option letters.
- You ONLY see candidate time windows with short captions (no raw frames). Use ONLY
these captions; do not invent details.
- Read the User question and extract minimal visual facts that would distinguish the
options (entities, actions, attributes, relations).
- Treat the Visual retrieval query as a short checklist, not a question.
- For each candidate window, judge relevance to the question based ONLY on its caption.

Task:
- For each window (in the given order), label it RELATED / PARTIAL / UNRELATED and
briefly explain why.
- Separate KNOWN vs UNKNOWN facts relative to the question/options; UNKNOWN should focus
on option-distinguishing facts that are missing or unclear.
- Propose concrete Next search cues (short phrases) that target the UNKNOWN facts. Do
NOT include timestamps or relative-time words.

Output format (NO JSON, keep this order):
Span relevance: <for each span, label RELATED / PARTIAL / UNRELATED with a short reason>
Evidence: <1-3 sentences; include KNOWN facts and UNKNOWN facts tied to the
question/options>
Next search: <3-8 short phrases, derived from UNKNOWN facts; same language as the
question when possible>
USEFUL_SPANS:
- HH:MM:SS{HH:MM:SS
- ... (up to {MAX_USEFUL} lines; must be chosen from the candidate list)

Notes:
- If none clearly match, say so in Evidence, but still list up to 1-2 least-irrelevant
spans in USEFUL_SPANS as weak leads.
- Do not add any extra sections beyond the required output lines and USEFUL_SPANS block.

User question (may include options; use them to disambiguate): {USER_QUESTION}
Visual retrieval query: {QUERY_TEXT}

Candidate windows:
1. [HH:MM:SS{HH:MM:SS] <caption>
2. [HH:MM:SS{HH:MM:SS] <caption>
...
```

## F.5. Prompt for Clip Captioning

---

**Clip Caption Prompt**

---

```
You are a vision-language assistant. You will be given multiple frames from a single
video clip spanning {START_HMS}{{END_HMS}.
Describe ONLY what is visible on screen (no audio/subtitles/metadata or outside
knowledge).
Write a detailed, chronological visual narrative that reads naturally as a story.

Return EXACTLY one JSON object (no extra text, no markdown) with this schema:
{
  "clip_description": "<multi-sentence (preferably multi-paragraph) narrative of the
visible events.
Be concrete: who/characters (use neutral labels if unknown), clothing/colors/patterns;
objects (names/quantities/states);
setting/location cues (indoor/outdoor/room/setting); camera movement; lighting; legible
on-screen text;
entrances/exits; occlusions; and state changes across the clip. Organize from start to
end;
you may use lightweight hints like [start]/[mid]/[end] or [t$\approx$HH:MM:SS] if
helpful|do not invent exact timestamps.
Strictly visual evidence only.>"
}

Writing requirements:
- Visual-only: avoid speculation, backstory, or intent beyond what is visibly evident.
- Coverage is within the prose (no lists): weave people, objects, scene, actions,
camera, lighting, on-screen text, and changes into a coherent narrative.
- Chronology: describe from the start toward the end; you may include lightweight hints
like [start], [mid], [end] or [t$\approx$00:00:05] if helpful|do not invent exact
timestamps.
- JSON constraint: return exactly one JSON object with the single key
'clip_description'; no code fences, no extra keys, no trailing text.
```

---

## F.6. Prompt for Trajectory-Level Semantic Groundedness

> **LLM as Judge Prompt**
>
> ```
> You are a rigorous auditor for Video Agent Trajectories.
> Your job: evaluate whether the agent's video-search process is logically grounded AND
> whether its trajectory is clear and traceable toward the final answer (agentic
> interpretability).
>
> ===============
> INPUTS
> ===============
> User Query:
> {{query}}
>
> Search Trajectory Log:
> {{trajectory_log}}
> Format: Turn # | Action | Tool Output | Agent Thought
>
> Final Answer:
> {{final_answer}}
>
> ===============
> CORE RULE: STRICT EVIDENTIARY SUPPORT
> ===============
> Let A be the critical Tool Output(s) provided during the search process.
> Let B be the claim in the Final Answer.
>
> The audit focuses strictly on whether A supports B.
> - If A supports B (explicitly contains, entails, or validates B), the answer is
> GROUNDED.
> - If B asserts claims, facts, or details not found in A, or contradicts A, the answer is
> HALLUCINATED.
>
> ===============
> FAIRNESS: FOCUS ON FINAL ANSWER
> ===============
> - **Scope:** Hallucination judgment applies ONLY to the {{final_answer}}.
> - **Thoughts Ignored:** Speculative, incorrect, or wandering statements inside "Agent
> Thought" do NOT count as hallucination unless those errors propagate into the Final
> Answer.
> - **Authority:** Regardless of whether the agent acts as a relay (copying a tool) or an
> analyst (synthesizing data), the rule remains: Does the tool evidence support the final
> claim?
>
> ===============
> METRICS (OBSERVABLE)
> ===============
>
> 1) Hallucination (hallucination: true/false)
>
> Definition: Hallucination is the absence of tool-based evidence for the Final Answer.
>
> Compare the Final Answer (B) against the visible Tool Outputs (A).
>
> **hallucination = false** (Not Hallucinated) IFF:
>   - The content of the Final Answer is explicitly present in or strictly entailed by the
> Tool Outputs.
>   - (For Multiple Choice) The option selected matches the evidence or the explicit
> answer provided by the tool.
>
> **hallucination = true** (Hallucinated) IFF:
>   - **Fabrication:** The Final Answer includes specific visual details, timestamps,
> counts, or objects that appear NOWHERE in the Tool Outputs.
> ```

  – **Contradiction:** The Final Answer claims "X" while the Tool Outputs explicitly say
"Not X" or "Y".
  – **Unjustified Certainty:** The Final Answer provides a definitive/specific answer
when the Tool Output was empty, ambiguous, or explicitly stated "I don't know" or "No
information found".

2) Trajectory-to-Answer Clarity (trajectory_clarity: 0--10)
How reconstructable and \toward-the-answer" the trajectory is (independent of
correctness).
Capability-adjusted scoring:

Score guidance:
10: Each major step states intent, cites prior evidence, refines spans/queries, and the
final answer is clearly linked to specific evidence.
7{9: Mostly coherent; occasional unclear jumps but overall traceable.
4{6: Fragmented; weak linkage between actions and evidence; rationale often implicit.
1{3: Largely opaque; actions feel arbitrary; hard to explain why the agent did what it
did.
0: No meaningful trajectory; cannot reconstruct a path to the answer from the log.

===============
CREDIBILITY SCORE (credibility_score: 0--10)
===============
Credibility measures the strength of the evidential support for the Final Answer.

10: Strong Support. The Tool Outputs contain the exact answer or decisive visual
evidence covering the full query scope.
8{9: Good Support. Minor ambiguity or slight coverage gap, but the answer is highly
likely based on tool data.
5{7: Weak/Partial Support. Evidence is indirect, circumstantial, or misses a key part of
the query (e.g., checking only half the video for a "first occurrence" question).
0{4: Unsupported/Hallucinated. The answer is based on no evidence or contradicted by
evidence.

===============
REQUIRED AUDIT STEPS (MUST DO)
===============
1) Identify the critical Tool Output(s) (A) that relate to the Final Answer (B).
2) Quote (verbatim, short snippet) the specific text in A that supports or contradicts
B.
3) CHECK SUPPORT:
   – Does A contain the information in B? -> hallucination = false.
   – Does B contain information absent from A? -> hallucination = true.
4) Rate trajectory_clarity based on logical progression.
5) Set credibility_score based on how strong that support is.

===============
OUTPUT (JSON ONLY)
===============
Return valid JSON exactly in this schema (no extra text):
{
  "reasoning": "Concise justification. 1) Quote the tool evidence. 2) State if Final
Answer matches/is supported by this evidence. 3) Explain scoring for
clarity/credibility.",
  "hallucination": true/false,
  "trajectory_clarity": 0-10,
  "credibility_score": 0-10
}

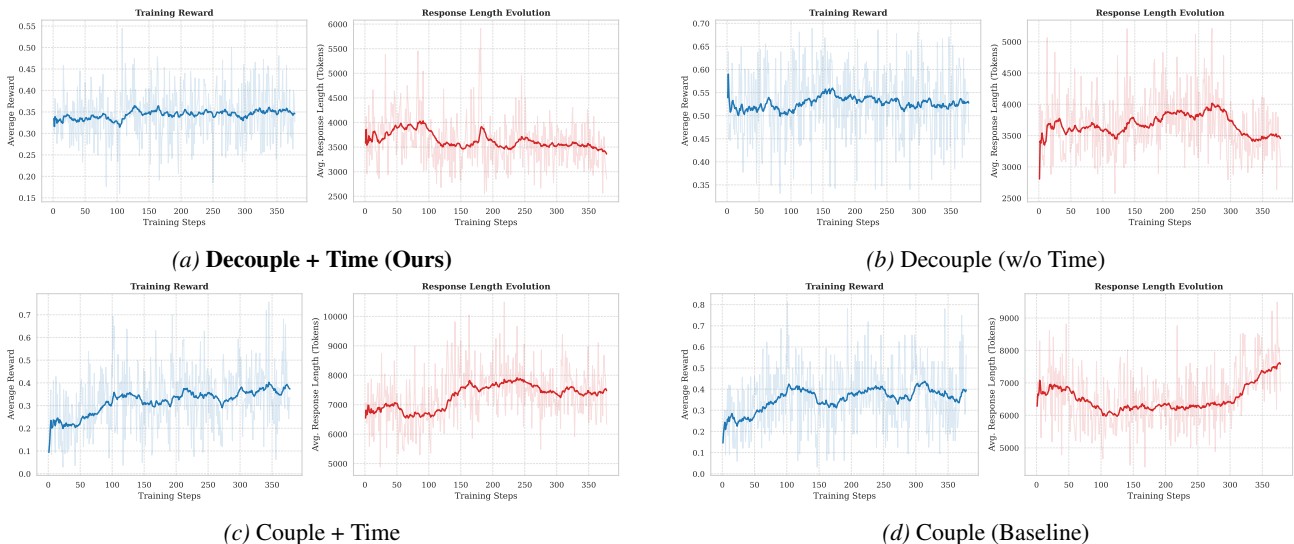

*(a)* **Decouple + Time (Ours)**                    *(b)* Decouple (w/o Time)

*(c)* Couple + Time                    *(d)* Couple (Baseline)

*Figure 13.* **Training Dynamics across Model Configurations.** We report the moving average of the reward (left y-axis) and response length (right y-axis) for four variants. The proposed *decoupled + time* setting achieves a superior balance between high reward attainment and concise tool usage.

# G. Training Dynamics of Agent Variants

To empirically validate our design, we study GRPO training stability and convergence under four configurations, contrasting our decoupled planner with coupled agent baselines, each instantiated with and without explicit temporal cues. Figure 13 reveals clear differences in optimization dynamics. The time-aware decoupled variant exhibits the smoothest learning curves: rewards increase steadily, and the interaction horizon remains well regulated rather than drifting toward longer, noisier trajectories. Removing temporal cues does not break stability, but it slows evidence localization, suggesting that temporal grounding primarily improves sample efficiency rather than acting as a stabilizer. In contrast, the coupled baselines show pronounced oscillation and high variance in both reward and trajectory length across training, consistent with optimization interference when navigation decisions and answer authority are optimized within a single shared history. By isolating the search policy from answer termination and enforcing an inspector-gated endpoint, our decoupled protocol reduces this interference and yields more stable RL fine-tuning.

