# OpenReview forum: "VideoSEAL: Mitigating Evidence Misalignment in Agentic Long Video Understanding by Decoupling Answer Authority"
_ICML.cc/2026/Conference — ICML 2026 regular_

### Official Review · Reviewer_4KAV · 2026-03-05

**Soundness:** 3
**Presentation:** 3
**Significance:** 2
**Originality:** 2
**Overall Recommendation:** 4
**Confidence:** 3

**Summary:**

This paper introduce a decoupled planner-inspector framework to address prompt pressure and reward pressure. A language model planner is responsible for long-horizon navigation and evidence seeking using tools. A separate frozen MLLM inspector then verifies the proposed visual evidence and holds exclusive authority to provide the final answer.

**Compliance With Llm Reviewing Policy:**

Affirmed.

**Final Justification:**

I appreciate the rebuttal and will maintain positive rating.

**Key Questions For Authors:**

1. Whether the decoupled method lead more inference cost ?
2. What is the key difference between VideoSEAL and Gens? And the performance comparison is needed.
3. Could you provide the training dynamics of planner?

**Limitations:**

Please include the potential limitations of decoupled architecture like additional inference cost.

**Strengths And Weaknesses:**

Strengths:
1. Simple and Effective Architectural Solution
2. The paper provides a comprehensive experimental evaluation across four challenging benchmarks (VideoMME, MLVU, LongVideoBench, LVBench).
3. Clear articulation and diagnosis of "evidence misalignment." The concepts of "prompt pressure" and "reward pressure" are intuitive and well-supported by diagnostic experiments

Weaknesses:
1. The core idea is similar to decoupling evidence seeking and question answering (e.g., Gens[1]), which weaken the novelty of the paper.
2. The inspector is implemented with Qwen-VL series,  what if using Kimi-VL and Intern-VL.
3. Missing some related works/baselines:
a) FrameThinker: Learning to Think with Long Videos via Multi-Turn Frame Spotlighting
b) Conan: Progressive Learning to Reason Like a Detective over Multi-Scale Visual Evidence
4. Inspector is a frozen, prompted MLLM; generalization and potential need for fine-tuning unexplored
5. Planner's iterative cost/ latency breakdown could be clearer

[1] GenS: Generative Frame Sampler for Long Video Understanding, 2025.

---

> ### Author Rebuttal · Authors · 2026-03-30
>
> We thank the reviewer for the constructive feedback and address the concerns below.
>
> >**W1/Q2: Relation to GenS and novelty**
>
> We acknowledge the resemblance, while GenS and VideoSEAL address different problems. **GenS** is a plug-and-play **generative frame sampler** that improves query-relevant span selection for a downstream VideoLLM. **VideoSEAL** instead targets **decision-stage evidence misalignment** in **agentic** LVU, where the same agent both plans and answers under prompt/reward pressure (Sec. 3.3; Fig. 2). Our key contribution is therefore **not span selection**, but **reassigning stop+final-answer authority** to a frozen inspector gate (Sec. 4.3; Tables 2–3), which can refuse and request another turn when inspected evidence is insufficient. The conceptual difference are illustrated in Tab R0 below.
>
> Under the same **Qwen2.5-VL-7B** backbone and fixed budget, VideoSEAL consistently outperforms a GenS baseline across benchmarks (Tab. R1). We will clarify the exact GenS setup in the revision.
>
> **Tab R0: GenS vs VideoSEAL (Conceptual)**
>
> |Method|Core insight|Insufficient->Refinement|Train method|Diagnostics|
> |:---|:---|:---:|:---|:---:|
> |GenS|Query-aware span selector|✗|SFT (sampler)|✗|
> |VideoSEAL|Evidence alignment|✓|GRPO (planner)|✓|
>
> **Tab R1: GenS vs VideoSEAL (Performance)**
>
> |Method|LongVideoBench|MLVU|LVBench|Video-MME (w/o sub, overall)|
> |:---|---:|---:|---:|---:|
> |GenS (Qwen2.5-VL-7B)|46.8|54.2|34.8|53.3|
> |Ours (Qwen2.5-VL-7B)|**62.0**|**68.2**|**55.1**|**62.9**|
>
> >**W2: Inspector backbone (Kimi/InternVL) choice**
>
> We agree that inspector choice is important. The inspector is **modular** and can be swapped at inference without retraining. Appendix Table 6 already evaluates **GPT-4o** and **Gemini-3-Flash-preview** in the scaling analysis, and shows the same pattern that stronger inspectors help more than scaling the planner. We now further add results with **Kimi2.5** under the same pipeline; see Reviewer **eWfT**, **Tab. R4**. The results show Kimi achieves much better absolute performance with our method, suggesting that the benefit is not tied to the original Qwen-VL choice. OpenRouter does not currently support InternVL. If support becomes available, we will include it in the revision.
>
>
>
> >**W3: Missing baselines (FrameThinker, Conan)**
>
> We agree these are relevant and will add them to the related work and experimental discussion. Conceptually, **FrameThinker** and **Conan** mainly strengthen **evidence selection/retrieval**, while our contribution operates at **answer authority and evidence alignment**; these are orthogonal and potentially composable. We include a comparison in Tab. R3, where VideoSEAL remains stronger on the reported benchmarks.
>
> **Tab R3: Comparison to FrameThinker/Conan**
>
> |Method|LongVideoBench|MLVU|LVBench|Video-MME (w/o sub, overall)|
> |---|---:|---:|---:|---:|
> |FrameThinker|52.9|59.1|36.6|-|
> |Conan|56.6|63.4|39.2|60.5|
> |Ours|**62.0**|**68.2**|**55.1**|**62.9**|
>
> >  **W4: Generalization and potential need for fine-tuning Inspector unexplored**
>
> We freeze the inspector deliberately to keep answer authority stable while optimizing only the planner, reducing co-adaptation and reward hacking. This is consistent with Fig. 7, where decoupling avoids the growing hallucination gap seen in coupled training. We will explore the training of both Inspector and Planner together in the future study.
>
>
>
> >**Q1/W5: Inference cost/latency**
>
> Yes, decoupling introduces an additional inspector step, so worst-case latency can increase. We already report overall efficiency in **Table 8**, where the decoupled system achieves the best accuracy in that comparison while also having the lowest average cost among the agentic baselines (**USD 0.015/query**). We further add **Tab. R4** with a stage-wise LVBench breakdown, showing that **inspection dominates latency** while **retrieval/filtering dominates cost**. We will include this clearer breakdown in the revision and explicitly list added latency as a limitation.
>
> **Tab R4: Cost/Latency Breakdown**
>
> |Stage|Model|Total Prompt Tokens|Prompt/sample|Total Completion Tokens|Completion/sample|Total Cost|Cost/sample|Cost Share|Total Latency|Latency/sample|Latency Share|
> |---|---|---:|---:|---:|---:|---:|---:|---:|---:|---:|---:|
> |retrieve|DeepSeek-V3.2 filter|29,196,682|19,045.5|1,244,168|811.6|15.925|0.010|69.25|28,872.369|18.834|27.10|
> |planner|Qwen3-8B|18,271,896|11,919.0|2,246,481|1,465.4|3.579|0.002|15.56|18,890.512|12.323|17.73|
> |inspect|Inspector|42,855,608|27,955.4|311,147|203.0|3.447|0.002|14.99|58,232.133|37.986|54.66|
> |fallback|Qwen3-8B|137,957|90.0|39,124|25.5|0.045|0.000|0.19|548.485|0.358|0.52|
>
> >**Q3: Planner training dynamics**
>
> Only the **planner** is updated with GRPO; the **inspector remains frozen**. Figs. 1/7/13  in the paper show stable training dynamics. Unlike coupled baselines that develop a widening hallucination gap, the decoupled setup keeps answer accuracy much better aligned with groundedness during training.

---

> > ### Author Rebuttal · Reviewer_4KAV · 2026-04-01
> >
> > The authors have provided detailed clarifications and additional experiments that address my concerns. I will consider raising my rating after further discussion with AC and other reviewers. In the revised manuscript, please ensure that the rebuttal content—particularly the comparisons with Gens, FrameThinker, and Conan—is properly incorporated.

---

> > > ### Author Response · Authors · 2026-04-01
> > >
> > > Thank you for the positive follow-up. We are glad our clarifications and additional experiments have addressed your concerns.
> > >
> > > As suggested, we will incorporate the rebuttal content into the revised manuscript (in the final version if accepted), in particular (i) the controlled comparison and discussion vs. **GenS** and (ii) the comparisons with **FrameThinker** and **Conan** (with proper citations and placement in the main text/experiments).
> > >
> > > We also look forward to feedback from the AC and the other reviewers, and are happy to clarify anything further.

---

### Official Review · Reviewer_3mJn · 2026-03-05

**Soundness:** 3
**Presentation:** 2
**Significance:** 3
**Originality:** 3
**Overall Recommendation:** 4
**Confidence:** 3

**Summary:**

This paper studies evidence misalignment in agentic long-video QA, where answers can be correct but insufficiently grounded in visual evidence. It proposes two diagnostics and a decoupled planner–inspector framework that separates evidence search from final answer authority. Results on multiple benchmarks show consistent improvements over a matched coupled baseline.

**Compliance With Llm Reviewing Policy:**

Affirmed.

**Key Questions For Authors:**

1. Can the authors better isolate the contribution of architectural decoupling from that of the auxiliary components (captioner, retriever/filter, inspector), e.g., through tighter controlled ablations?

2. How reliable is the semantic groundedness metric in practice? Is there any human evaluation or agreement analysis to validate the LLM-judge decisions?

3. The paper argues that stronger inspectors and larger budgets help, but can the authors clarify when these gains saturate and whether the same trend holds uniformly across benchmarks?

**Limitations:**

The paper includes a impact statement, but a more explicit discussion of method limitations and potential negative societal impacts would strengthen the submission.

**Strengths And Weaknesses:**

Strengths.

* The paper addresses an important and realistic failure mode in long-video agents: correct but weakly grounded answers. The proposed diagnostics are intuitive and help make the problem more explicit.
* The planner–inspector decoupling is a clean and practically meaningful design; separating search from answer authority is a sensible architectural intervention.
* The empirical section is reasonably comprehensive, including main comparisons, reward ablations, search scaling, inspector scaling, and refusal analysis.

Weaknesses.

* The main novelty is more system-level than algorithmically deep; the method largely combines known components with a decoupled control design.
* Some core claims rely heavily on the proposed diagnostics, especially the LLM-judge-based semantic groundedness, whose reliability is not fully validated.

---

> ### Author Rebuttal · Authors · 2026-03-30
>
> We thank the reviewer for the positive assessment and thoughtful questions.
>
> >**W1: System-level novelty**
>
> We agree that the contribution is more *structural/system-level* than a new learning algorithm. Our novelty is to identify a specific failure mode in agentic LVU, *decision-stage evidence misalignment* under prompt/reward pressure, and address it with a minimal but consequential change in ***answer authority***. In the standard coupled setup, the same agent both searches and commits to the final answer. Whereas, in VideoSEAL, the planner performs evidence seeking, while the inspector gate alone holds stop+final-answer authority (Sec. 4.3). We intentionally keep the captioner/retriever/filter standard. Table 2 is the key matched control: tools, backbone, and objective are fixed, and the gain persists under both rewards, indicating that the improvement is primarily due to decoupling, not auxiliary modules. Table 3 further supports the intended behavioral shift toward reduced forced guessing. We will foreground this contribution more clearly in the revision.
>
>
>
> >**W2 & Q2: Reliability of semantic groundedness (LLM judge compared to human evaluation)**
>
> We treat $G_{s}/H_{s}$ (Eq. 3-4; Table 4) as a *comparative* diagnostic for correct-but-unsupported answers, complementary to temporal groundedness  ($G_{t}/H_{t}$; Eq. 1; Table 2). The motivation is that an agent may answer depending toward unsupported evidence fitting (while degrading $G_s$), even access to relevant content (so $G_t$ saturates).
>
> To validate the LLM judge, we additionally collected labels from **5 human experts** on **200 LVBench trajectories**.
>
> The LLM judge we use achieves **93.7%** hallucination-detection accuracy ($\kappa$=0.774, F1=0.813). Its clarity/credibility also correlate well with humans (Spearman $\rho = 0.766/0.834$, QWK = 0.564/0.821), with **91.5% / 80.4%** of predictions within $\pm$ 1. These results support  $G_{s}/H_{s}$  as a reliable *comparative metric*, though not as an oracle. We will add the protocol, prompts, and validation details in the revision.
>
>
>
> >**Q1: Isolating architectural decoupling vs auxiliary components**
>
> We agree this should be clearer. Table 2 provides the main controlled ablation: with the **backbone, objective, and tools fixed**, the decoupled design consistently improves over the coupled design. We further add Tab. R1, where the decoupled architecture is held fixed while swapping auxiliary components. On LVBench, changing the captioner, retriever, or LLM filter changes performance by only $\le$ 1.2, much smaller than the **coupled→decoupled** gap. This suggests that *decoupling is the main source of gain*, while auxiliary modules provide smaller refinements. We will clarify this decomposition in the revision.
>
> **Tab R1: Auxiliary component swaps (decoupled fixed)**
>
> |Module|Setting (decoupled fixed)|LVBench|
> |:---|:---|:---:|
> |Captioner|Qwen3-VL-8B-Instruct|55.1|
> |Captioner|Gemini-3-Flash-preview|56.2|
> |Retriever|text-embedding-3-large|55.1|
> |Retriever|BM25|53.6|
> |LLM filter|DeepSeek-V3.2|55.1|
> |LLM filter|GPT-4o|55.0|
>
> We also clarify that the inspector is not only a post-hoc verifier. As described in Sec. 4.3 and Fig. 10, it has answer and stop authority through the inspector gate. Inspector takes only $q$ and $v_{t}$ as input and can return either an answer or an insufficient-evidence verdict with feedback.
>
>
>
> >**Q3: Saturation of stronger inspectors / larger budgets**
>
> We appreciate this question. The budget analysis is now summarized in **Tab. R2**. Across benchmarks, gains come mainly in the **early turns** and then largely saturate: e.g., LVBench improves from **47.1→55.1** by **16** turns and is essentially flat afterward; Video-MME slightly declines after 16; MLVU and LongVideoBench continue improving only marginally. Thus, the trend is consistent in direction, though the exact saturation point varies by benchmark. For stronger inspectors, the controlled evidence is currently strongest on LVBench: **Fig. 6(b)** shows monotonic gains within the Qwen2.5-VL family, and **Appendix Table 6** shows the same qualitative trend with stronger visual backbones. We will make these two claims more clearly separated in the revision.
>
> **Tab R2: Interaction depth $T$ and performance vs budget $K$ (turns)**
>
> |Benchmark|Avg $T $ when  $K =32$|$K=4$|8|12|16|20|24|28|32|
> |:---|:---:|:---:|:---:|:---:|:---:|:---:|:---:|:---:|:---:|
> |MLVU|3.9|63.9|65.5|66.9|68.2|68.5|68.6|68.6|68.7|
> |Video-MME (w/o sub)|3.8|58.4|60.1|61.6|62.9|62.8|62.8|62.7|62.7|
> |LongVideoBench|5.2|55.3|57.8|60.1|62.0|62.1|62.2|62.1|62.1|
> |LVBench|4.1|47.1|50.5|52.8|55.1|55.2|55.1|55.1|55.1|
>
> >**Limitations**
>
> We agree this should be made more explicit. VideoSEAL reduces but does not eliminate false-positive sufficiency verdicts, and it adds inference cost/latency. We will expand the limitations discussion.

---

> > ### Author Rebuttal · Reviewer_3mJn · 2026-04-04
> >
> > Thanks the authors for the clarifications. The rebuttal addressed my main concerns.

---

> > > ### Author Response · Authors · 2026-04-04
> > >
> > > We thank the reviewer for the careful follow-up and for confirming that the rebuttal addressed the main concerns. We will incorporate these clarifications in our revision, including the validation details for the semantic groundedness metric, the controlled ablations, and the expanded discussion of limitations.

---

### Official Review · Reviewer_d27r · 2026-03-09

**Soundness:** 3
**Presentation:** 2
**Significance:** 3
**Originality:** 2
**Overall Recommendation:** 4
**Confidence:** 3

**Summary:**

This paper studies the problem of evidence misalignment in long video understanding (LVU), where the model can generate the correct answer, but the retrieved evidence or reasoning trajectory cannot truly support the answer. To analyze this phenomenon, the authors propose two metrics, temporal groundedness and semantic groundedness, to evaluate the evidence alignment in the reasoning process.
Based on the above analysis, the paper proposes the VideoSEAL framework, which decouples the traditional agent architecture into two modules: planner and inspector.  Experimental results show that the decoupled architecture achieves certain performance improvements compared with the coupled architecture on several long video understanding benchmarks.

**Compliance With Llm Reviewing Policy:**

Affirmed.

**Final Justification:**

The response has basically solved my concerns. I have increase my rating to 4

**Key Questions For Authors:**

1. The inspector module proposed in the paper is functionally similar to verifier-based agent architectures (such as VideoMind). Can the authors further clarify the differences between this method and these existing frameworks in terms of design ideas and technical implementation?
2. If the inspector incorrectly judges that the evidence is sufficient or insufficient, what impact will it have on the overall reasoning process? Does the system have certain robustness to mitigate this situation?
3. Have more fine-grained ablation experiments been conducted to analyze the independent contribution of the planner–inspector decoupling itself compared with other factors (such as reward design or retrieval strategies)?

**Limitations:**

yes

**Strengths And Weaknesses:**

# strength

1. Temporal groundedness and semantic groundedness provide an operational tool for analyzing whether the model's reasoning trajectory is truly based on video evidence.
2. Separating evidence search and final answer generation into two roles, planner and inspector, is a clear and easy-to-understand structural design.
3. The experimental results show certain consistency. On multiple long video understanding benchmarks, the decoupled architecture achieves certain performance improvements under the same backbone.

# weakness

1. Although the paper proposes groundedness metrics, the experimental section still mainly focuses on final answer accuracy, and there is relatively little detailed analysis of reasoning trajectory quality, error types, or evidence alignment. In addition, the independent contribution of some key components (such as inspector gating) could be further verified through more systematic ablation.
2. The inspector module proposed in the paper is used to determine whether the evidence is sufficient and to control the final answer generation, which is conceptually similar to existing verifier-based agent frameworks (such as VideoMind). These methods also introduce additional modules to verify reasoning or answers. The paper does not sufficiently explain the differences and new contributions compared with these methods.

---

> ### Author Rebuttal · Authors · 2026-03-30
>
> We thank the reviewer for the careful reading and constructive feedback.
>
> >**W1: Lack of detailed analysis on trajectory quality, error types, and systematic ablation of inspector gating**
>
> We agree these analyses should be more explicit. We have added consolidated grounding and trajectory-quality results in Reviewer **eWfT**, W1, **Tab. R1**.
>
> In brief, on LVBench, VideoSEAL reduces **correct-but-unsupported** answers and improves credibility; the main residual errors are **fine-grained perception failures** and **off-target retrieval / coverage misses** (Appendix D, Fig. 11–12). We will move this summary into the main paper.
>
> For the independent contribution of inspector gating,  **Table 2** is the key controlled ablation: coupled vs. decoupled under the **same backbone, toolchain, budget, and reward regime**, differing only in whether **answer+stop authority** is assigned to the inspector. The gain persists under both reward settings. **Table 3** further probes the gate itself by comparing refusal and answer behavior on **non-target** vs. **ground-truth** clips. We will clarify this ablation path more explicitly in the revision.
>
>
>
> >**W2 & Q1: Differences and new contributions compared to verifier-based agent frameworks (e.g., VideoMind)**
>
> We appreciate this question and believe the distinction should be made clearer: VideoSEAL $\\underline{\\textbf{is not a verifier added to re-rank candidate spans.}}$ Its key contribution is a **structural reassignment of answer authority** to reduce evidence misalignment (correct-but-ungrounded answer) under prompt/reward pressure.
>
> - **VideoMind-style frameworks**: a role-based pipeline (Planner→Grounder→Verifier→Answerer) where the Grounder proposes multiple candidate temporal spans and the Verifier **selects** the most reliable one .
> - **VideoSEAL (ours)** instead decouples evidence seeking from final answering: the planner $P$ **seek/select/refine** spans all on itself, while the inspector $I$ alone holds **stop+final-answer authority** and tell $P$ whether the currently inspected evidence can support for answering $q$. $I$'s answer is conditioned only on $q$ and the *currently* selected clip $v_t$ from $P$. $P$ refine/reselect the span on the feedback from $I$, rather than using $I$ to select a good span.
>
> This also leads to more refinement and produce more accurate evidence: compared with VideoMind, VideoSEAL shows stronger refusal on non-target clips and stronger recovery / post-hit lock-in in the trajectory statistics (Reviewer **eWfT**, W3, **Tab. R3**).
>
> **Tab R2: Comparison of VideoMind vs VideoSEAL**
>
> | Method | Core insight | Insufficient->Refinement | Train method | Additional verifier | Diagnostics |
> | :--- | :---: | :---: | :---: | :---: | :---: |
> | VideoMind | Span selector | ✗ | SFT | ✓ | ✗ |
> | VideoSEAL (Ours) | Evidence alignment | ✓ | GRPO | ✗ | ✓ |
>
>
>
> >**Q2: Robustness to two types of incorrect inspector verdicts**
>
> We agree robustness to incorrect verdicts is important. The two error types have asymmetric effects. Incorrectly judging that the evidence is sufficient is *more harmful* because it terminate the trajectory early on unsupported evidence. Incorrectly judging that the evidence is insufficient is *less harmful*: it mainly costs a few extra search steps, and can often be recovered in the closed loop through inspector feedback (Fig. 10).
>
> In fact, according to results from Table 3 in the paper, the inspector $I$ is much more likely to refuse on non-target clips than on ground-truth clips, and much likey to answer on ground-truth clips than non-target clips, which reduces harmful premature stopping. With a stronger frozen inspector, this pattern becomes even sharper. We will clarify this asymmetry and the corresponding mitigation mechanisms in the revision.
>
> >**Q3: Have fine-grained ablations on the independent contribution of planner–inspector decoupling?**
>
> Yes. **Table 2** is precisely the controlled ablation for the independent contribution of decoupling: the backbone, retrieval stack, filter, objective, and budget are matched, and only **answer+stop authority** changes. The improvement is consistent on LVBench and also appears on LongVideoBench.
>
>  Fig. 6  also shows that the coupled variant degrades at longer horizons whereas the decoupled variant continues to improve. Since only the planner is updated during GRPO and the inspector remains frozen, the training setup further isolates the effect of **authority decoupling** rather than conflating it with joint optimization.
>
> Overall, we appreciate the reviewer’s concerns and will revise the paper to make the added trajectory analysis, verifier distinction, robustness discussion, and ablation logic more explicit in the main text.

---

> > ### Author Rebuttal · Reviewer_d27r · 2026-04-05
> >
> > The response has basically solved my concerns. I have increase my rating to 4

---

> > > ### Author Response · Authors · 2026-04-06
> > >
> > > Thank you for the follow-up and for the updated rating. We appreciate that our response has **basically addressed your concerns**.
> > >
> > > The issues you pointed out are important, especially the grounding / trajectory analysis, the role of planner-inspector decoupling itself, the robustness of inspector decisions, and the distinction from verifier-based frameworks such as VideoMind. We will **revise these parts accordingly** so that they are **reflected more directly** in the paper.
> > >
> > > If there are any further follow-up questions, we would be happy to clarify them.

---

### Official Review · Reviewer_eWfT · 2026-03-13

**Soundness:** 2
**Presentation:** 3
**Significance:** 3
**Originality:** 2
**Overall Recommendation:** 4
**Confidence:** 4

**Summary:**

This paper addresses the “evidence mismatch” problem in agent-based long video understanding, where agents give correct answers without evidence in their reasoning traces. It proposes VideoSEAL, a decoupled planner–checker framework. The planner searches for evidence, while a frozen MLLM checker has sole authority to generate the final answer after verifying sufficient evidence. Experiments on benchmarks such as LVBench and Long VideoBench show improved accuracy and evidence alignment, and the system scales easily by upgrading the checker or increasing search budget without retraining the planner.

**Compliance With Llm Reviewing Policy:**

Affirmed.

**Final Justification:**

The additional experiments improve clarity and address several of my concerns. However, attribution of gains and presentation of key metrics could still be clearer. Overall, my stance is more positive than before, so I raise my rating.

**Key Questions For Authors:**

Weakness 1 and 3 are key questions for authors. I will raise the rating if they have reasonable answers.

**Limitations:**

yes

**Strengths And Weaknesses:**

Strengths
1. Innovative diagnostic tool: The paper systematically defines the phenomenon of Evidence Misalignment and introduces temporal and semantic groundedness metrics to reveal the structural causes of hallucinations in agent models for long-video understanding.

2. Efficient decoupled architecture: The proposed Planner–Inspector framework separates planning from answering and introduces a pixel-level verification “inspector gate,” alleviating prompt pressure during inference and reward pressure during training.

3. Strong scalability and performance: Experiments show significant improvements across multiple long-video benchmarks, and the framework enables smooth performance scaling by upgrading the visual backbone without retraining the planner.

Weaknesses
1. Absence of Quantitative Grounding Evaluation
While the paper introduces "temporal groundedness" and "semantic groundedness" as key diagnostics for evidence misalignment , the primary experimental results focus almost exclusively on answer accuracy (Table 1). The evaluation lacks a formal benchmark comparison of grounding accuracy (e.g., temporal IoU with ground-truth evidence) against existing agentic baselines to empirically prove that the framework actually yields more precise localization.

2. Fairness in Model Comparison and Training
The framework's performance gains are difficult to isolate due to several confounding factors in the experimental setup:
- Architectural Asymmetry: The system functions as an ensemble of a Planner and a specialized Inspector, making direct comparisons with monolithic, single-model agents inherently biased toward the increased parameter count.
- Supervision Disparity: The Planner is trained on CG-Bench, which provides dense, manually curated "clue" annotations. This provides a significant advantage in learning temporal localization compared to baselines trained without such granular grounding labels.
- Efficiency Metrics: A rigorous comparison requires a standardized analysis of inference efficiency to account for the overhead of the multi-model pipeline.

3. Lack of Behavioral Statistics and Trajectory Analysis
The paper fails to provide a statistical overview of the agent's actual operational behavior during inference:
- Interaction Depth: There is no data showing the average number of evidence retrieval turns required to reach a sufficiency verdict across different benchmarks.
- Strategy Characterization: It is unclear whether the learned behavior follows a coarse-to-fine refinement of a single temporal window or a stochastic search that frequently switches between disparate retrieval targets.

4. Utilization of Outdated Model Backbones
The empirical results rely on MLLMs and LLMs that are no longer represent the state-of-the-art in vision-language understanding. Given the rapid iteration in VLM capabilities, the current findings need validation with the latest frontier models (e.g., Gemini 3 or the newest Qwen series) to ensure the proposed structural decoupling remains necessary as model reasoning capacity improves.

---

> ### Author Rebuttal · Authors · 2026-03-30
>
> Appreciate the careful review. Point-by-point below.
>
> > **W1: Absence of Quantitative Grounding Evaluation**
>
> The grounding diagnostics evaluation are in the Table 4 of paper appendix, we apologize for not puting them in the main text. Tab R1 below puts them side-by-side on LVBench under the same setting: beyond higher accuracy, VideoSEAL yield significantly more groundness (IoU, $G_t$, $G_s$ ) and much less correct-but-ungrounded answers ($H_t$, $H_s$ ). In the revision, we will surface Tab R1 in the main text.
>
> **Tab R1: Grounding Evaluation on LVBench**
>
> ||tIoU_max|Recall@0.05 (= $G_{t,γ=0.05}$)|$G_{t,γ=0.1}$|$G_{t,γ=0.2}$|$H_{t}$ ↓||$G_{s}$ ↑|$H_{s}$ ↓|
> |:---|:---:|:---:|:---:|:---:|:---:|:---:|:---:|:---:|
> |LongVT|0.025|0.088|0.054|0.041|0.889||0.250|0.708|
> |VideoMTR|0.048|0.175|0.138|0.088|0.833||0.280|0.621|
> |VideoMind|0.098|0.275|0.219|0.144|0.629||0.273|0.543|
> |VideoAgent|0.124|0.281|0.231|0.169|0.838||0.454|0.471|
> |DrVideo|0.151|0.448|0.324|0.271|0.547||0.472|0.414|
> |Ego-R1|0.181|0.419|0.325|0.275|0.504||0.405|0.378|
> |**Ours**|**0.203**|**0.528**|**0.434**|**0.333**|**0.406**||**0.808**|**0.113**|
>
> *Takeaway: **VideoSEAL is not only more accurate, but more evidence-grounded**.*
>
>
> > **W2: Fairness in Model Comparison and Training**
>
> - **Architecture:** We agree this is an important concern. To isolate the effect of decoupling, **Table 2** is a matched control: the **planner, training, data, and tools are fixed**, the **inspector is frozen**, and only the **answer/stop authority** changes. The gap persists under both rewards, suggesting that the gain is not explained solely by extra parameters or ensemble structure.
> - **Supervision:** The gain is also not reducible to “dense localization supervision vs. none.” While our planner uses CG-Bench, several baselines (**Ego-R1 / LongVT / Video-MTR**) likewise use temporal labels in their own training.
> - **Efficiency:** Tables 8-9 report cost/latency/frames under the same budget; for a clearer stage-wise breakdown, see $\\underline{\\textbf{Review 4KAV, Tab R4}}$.
>
> *Takeaway: under matched training, decoupling is the driver; overhead is reported.*
>
>
> > **W3: Lack of Behavioral Statistics and Trajectory Analysis**
>
> - **Interaction Depth:** Please see $\\underline{\\textbf{Review 3mJn, Tab R2}}$. For **our method**, the average interaction depth across benchmarks is only **3.8–5.2** turns even with budge $K=32$, and performance gains arise mainly in the early turns before saturating (e.g., LVBench **47.1→55.1** by 16 turns). This indicates our gains came from more accurate evidence: most are resolved within a few turns, harder ones benefit from more evidence gathering.
> - **Strategy Characterization:** Tab. **R3** suggests retargeting then local refinement, not random switching: VideoSEAL hits evidence earlier (Hit@1/2/3), recovers more often after an initial miss (Recovery), and by step 2 moves closer to the correct span (Step2Closer). After the first hit, overlap rises and jumps shrink, and later steps stay near the evidence and refine locally (IoU_med, Jump_med, PostHitLockIn, PostHitLocalRefine).
>
> **Tab R3: Trajectory-Policy Statistics on LVBench**
>
> Defs: Hit@k=whether evidence is reached by step k; Recovery=whether a step-1 miss is corrected later; Step2Closer=whether step 2 moves closer to the correct span than step 1; IoU_med/Jump_med (pre→post)=after the first hit, overlap increases and jumps shrink; PostHitLockIn/PostHitLocalRefine=after the first hit, later steps stay near the evidence and refine locally.
>
> |Method|Hit@1|Hit@2|Hit@3|Recovery|Step2Closer|IoU_med|Jump_med|PostHitLockIn|PostHitLocalRefine|
> |:---|:---:|:---:|:---:|:---:|:---:|:---:|:---:|:---:|:---:|
> |Ego-R1|23.6|26.7|27.3|7.5|47.0|0.0→0.0|30s→22s|25.6|0.9|
> |DrVideo|25.5|34.8|39.8|33.0|45.3|0.0→0.0|512s→304s|13.0|0.0|
> |VideoMind|31.1|39.1|41.0|24.3|50.6|0.0→0.1|137s→134s|12.7|6.1|
> |Video-MTR|32.9|32.9|32.9|0.0|25.0|0.0→0.1|0s→976s|0.0|0.0|
> |**Ours**|**53.9**|**56.5**|**56.9**|**34.8**|**51.4**|**0.2→0.4**|**334s→64s**|**28.9**|**10.6**|
>
> *Takeaway: gains come from a **more effective evidence-seeking policy**, not longer traces.*
>
>
>
> > **W4: Utilization of Outdated Model Backbones**
>
> We agree that validation on newer frontier backbones is important. Appendix Table 6 already evaluates **GPT-4o** and **Gemini-3-Flash-preview** in the scaling analysis, and the same pattern holds there: scaling the inspector helps more than scaling the planner. We now add Tab R4 with end-to-end results for **Qwen3-VL-8B**, **Kimi2.5**, and **Gemini-3-Flash-preview** under the same pipeline across benchmarks.
>
> **Tab R4: Newer-Backbone Results and Inspector Modularity**
>
> |Inspector|LongVideoBench|LVBench|MLVU|Video-MME|
> |:---|:---:|:---:|:---:|:---:|
> |Qwen2.5 7B|62.0|55.1|68.2|62.9|
> |Qwen3 8B|69.2|58.5|79.2|74.0|
> |Kimi2.5|80.8|65.7|84.3|82.5|
> |Gemini-3-Flash-preview|77.2|69.9|79.3|83.0|
>
> *Takeaway: stronger backbones improve absolute performance, but decoupling remains beneficial under the same pipeline.*

---

> > ### Author Rebuttal · Reviewer_eWfT · 2026-04-04
> >
> > The additional experiments improve clarity and address several of my concerns. However, attribution of gains and presentation of key metrics could still be clearer. Overall, my stance is more positive than before, so I raise my rating.

---

> > > ### Author Response · Authors · 2026-04-05
> > >
> > > Thank you for the follow-up and for the more positive assessment. We appreciate that the additional experiments and clarifications were helpful, and we will incorporate them in the revision.
> > >
> > > On the remaining point about attribution of gains, the main evidence supporting our core contribution is the **matched-control comparison in Table 2**. This is the comparison that most directly isolates the effect of **decoupling planning from final answering**. The retrieval filter is helpful as well, but secondary, and we view its role mainly as reducing noisy context before it reaches the planner, consistent with our prompt-pressure analysis. We will make the relative roles of these components more explicit in the revision.
> > >
> > > For the **key metrics**, the main paper already defines the core diagnostics, while the rebuttal added a consolidated grounding comparison and additional trajectory statistics to directly address the reviewer’s questions on quantitative grounding evaluation and behavioral analysis. Because of the character limit, these additions had to be presented in a compact form. In the revision, we will **integrate them into the paper and define them more fully**, so that their connection to the grounding evidence and the architectural argument is clearer.

---

### Decision · Program_Chairs · 2026-04-30

**Decision:**

Accept (regular)

**Comment:**

The reviewers agreed that the paper tackles an important and practically problem: evidence misalignment in agentic long video understanding, and introduces a clear and intuitive decoupled planner–inspector framework.

Initially, reviewers have raised several concerns. First, the novelty is viewed as primarily system-level, with limited algorithmic innovation, and the relationship to prior verifier-based or decoupled frameworks is not always clearly distinguished. Second, the evaluation initially emphasized answer accuracy, with insufficient focus on grounding quality, trajectory behavior, and component-level attribution. Third, concerns were raised about fairness of comparisons (e.g., additional supervision, multi-module architecture), reliance on somewhat outdated backbones, and limited analysis of efficiency, inference cost, and behavioral dynamics.

The rebuttal substantially addressed many of these concerns. The authors provided additional grounding evaluations, trajectory statistics, controlled ablations isolating the effect of decoupling, and experiments on newer backbones. These clarifications improved confidence that the observed gains stem from the proposed architectural decoupling rather than confounding factors, and multiple reviewers acknowledged that their concerns were largely resolved or mitigated.

Overall, the paper presents a technically sound and well-motivated contribution, offering a useful perspective on agent design for long-horizon multimodal reasoning. While some limitations remain, the proposed decoupling paradigm is likely to inspire further work.

Therefore, I recommend a weak accept.